# Placental multi-omics integration identifies candidate functional genes for birthweight

Fasil Tekola-Ayele [1✉], Xuehuo Zeng[1], Suvo Chatterjee[1], Marion Ouidir[1], Corina Lesseur [2], Ke Hao [3], Jia Chen[2], Markos Tesfaye[4], Carmen J. Marsit [5], Tsegaselassie Workalemahu[6] & Ronald Wapner[7]

Abnormal birthweight is associated with increased risk for cardiometabolic diseases in later life. Although the placenta is critical to fetal development and later life health, it has not been integrated into largescale functional genomics initiatives, and mechanisms of birthweight-associated variants identified by genome wide association studies (GWAS) are unclear. The goal of this study is to provide functional mechanistic insight into the causal pathway from a genetic variant to birthweight by integrating placental methylation and gene expression with established GWAS loci for birthweight. We identify placental DNA methylation and gene expression targets for several birthweight GWAS loci. The target genes are broadly enriched in cardiometabolic, immune response, and hormonal pathways. We find that methylation causally influences *WNT3A*, *CTDNEP1*, and *RANBP2* expression in placenta. Multi-trait colocalization identifies *PLEKHA1*, *FES*, *CTDNEP1*, and *PRMT7* as likely functional effector genes. These findings reveal candidate functional pathways that underpin the genetic regulation of birthweight via placental epigenetic and transcriptomic mechanisms. Clinical trial registration; ClinicalTrials.gov, NCT00912132.

[1] Division of Population Health Research, Division of Intramural Research, Eunice Kennedy Shriver National Institute of Child Health and Human Development, National Institutes of Health, Bethesda, MD, USA. [2] Department of Environmental Medicine and Public Health, Icahn School of Medicine at Mount Sinai, New York, NY, USA. [3] Department of Genetics and Genomic Sciences, Icahn School of Medicine at Mount Sinai, New York, NY, USA. [4] Section of Sensory Science and Metabolism (SenSMet), National Institute on Alcohol Abuse and Alcoholism & National Institute of Nursing Research, National Institutes of Health, Bethesda, MD, USA. [5] Gangarosa Department of Environmental Health, Rollins School of Public Health of Emory University, Atlanta, GA, USA. [6] Department of Obstetrics and Gynecology, Maternal-Fetal Medicine, University of Utah, Salt Lake City, UT, USA. [7] Department of Obstetrics and Gynecology, Columbia University, New York, NY, USA. ✉email: ayeleft@mail.nih.gov

ndividuals with birthweight in the lower or upper bounds of the population distribution are at greater risk of cardiovascular and metabolic diseases during adulthood and mortality during childhood[1–5]. Therefore, advances in understanding the biological underpinnings of birthweight will have broad relevance to understanding the etiology of complex diseases and developing therapeutics. Genome-wide association studies (GWAS) have identified several genetic variants associated with fetal growth and birthweight[6–12]. However, most of the GWAS variants map to noncoding regions of the genome, and the functional mechanism through which the variants influence birthweight remains unclear.

Multi-omic molecular features of the placenta, one of the most functionally-relevant organs for fetal growth[13], can provide a remarkable opportunity to investigate the biological embedding of genetic influences on birthweight. Despite growing evidence supporting the key role of the placenta in regulating fetal growth and subsequent health, data on placental multi-omic molecular features integrated with genetic variation are rare[14]. Large-scale tissue-specific gene expression regulation resources such as the Genotype-Tissue Expression Portal (GTEx) do not include placenta[15]. As a result, birthweight GWAS signals that may be explained by regulatory effects on placental epigenome and transcriptome are not understood. Establishing that a variant driving the GWAS signal also influences nearby gene expression (i.e., expression quantitative trait locus, *cis*-eQTL) in a relevant tissue suggests a putative regulatory mechanism[16]. Recent work has indicated that integrating additional information from DNA methylation (DNAm) quantitative trait locus (*cis*-mQTL) strengthens the evidence for causality[17–19]. Therefore, the prioritization of candidate functional genes for birthweight can be facilitated by elucidating the coordinated relationships of GWAS variants with placental methylation and gene expression.

In this study, we provide functional mechanistic insight into birthweight by integrating GWAS variants known to be associated with birthweight with placental methylation and gene expression. Our primary analyses integrate placental DNAm, gene expression, and genotype data from participants of the *Eunice Kennedy Shriver* National Institute of Child Health and Human Development (NICHD) Fetal Growth Studies–Singleton cohort[20,21] with GWAS summary statistics of birthweight[6]. First, we determine whether birthweight GWAS variants regulate nearby gene expression (i.e., *cis*-eQTL) and DNAm (i.e., *cis*-mQTL) in the placenta, and assess functional features of the regulated genes. Second, for birthweight GWAS variants found to be both *cis*-eQTL and *cis*-mQTL, we elucidate the directional relationships among the genetic variants, DNAm and gene expression in the placenta using mediation[22] and Mendelian Randomization[23] approaches. Third, we apply multitrait colocalization[17] at each GWAS locus found to be both *cis*-eQTL and *cis*-mQTL to investigate whether all three traits (i.e., birthweight, placental DNAm, and gene expression) share the same causal variant and identify candidate genes for functional follow-up. Loci found to be both *cis*-eQTL and *cis*-mQTL are further assessed using an independent dataset from the Rhode Island Child Health Study (RICHS)[24].

## Results

**Overview of the analysis workflow.** An overview of the analysis workflow is presented in Fig. 1. *cis*-eQTL and *cis*-mQTL loci were identified using the NICHD Fetal Growth Studies–Singleton dataset. A total of 291 placental samples with genotype and DNAm data were included in *cis*-mQTL analysis. Samples were obtained from pregnant women with self-identified Hispanic ($n = 97$), White ($n = 74$), Black ($n = 71$), and Asian race/ethnicity

($n = 49$); mean age of 27.8 years; mean pre-pregnancy body mass index (BMI) of 25.1 kg/m$^2$; and 46.5% female infants. A subset of 71 samples with genotype and RNA-seq data (22 Hispanic, 21 White, 20 Black, 8 Asian) were included in *cis*-eQTL analysis. The characteristics of the study participants including maternal age, race/ethnicity, pre-pregnancy BMI, gestational duration, parity, and birthweight showed no significant difference between the participants with DNAm data and the subset with RNA-seq data (Supplementary Data 1). Out of 286 single-nucleotide polymorphisms (SNPs) found to be associated with birthweight at $P < 5 \times 10^{-8}$ (i.e., 190 lead SNPs with $P < 6.6 \times 10^{-9}$ and 96 SNPs with $6.6 \times 10^{-9} < P < 5 \times 10^{-8}$) by Warrington et al.[6], 273 SNPs were available in our dataset (Supplementary Data 2). A total of 7901 unique protein-coding gene mRNAs and 104,053 CpG DNAm sites found within 1 Mb distance from the 273 SNPs were included in eQTL and mQTL analyses, respectively. For loci found to be both eQTL and mQTL, we performed mediation, Mendelian Randomization, and multitrait colocalization analyses. Further evaluation of the co-occurring eQTL and mQTL associations was performed in RICHS ($n = 159$; 77.4% White; mean maternal age of 30.9 years; mean pre-pregnancy BMI of 26.6 kg/m$^2$; 49.7% female infants (Supplementary Data 1)). Functional annotation, regulatory enrichment, and pathway analyses were performed on all eQTL and mQTL target genes.

**_cis_-eQTLs and _cis_-mQTLs in the placenta for birthweight GWAS loci.** At a false discovery rate (FDR) adjusted $P < 0.05$, we identified 32 significant eQTL-eGene associations consisting 26/273 eQTL SNPs (vs. 0.04% expected genome-wide[25], $P = 1.97 \times 10^{-54}$, hypergeometric test) associated with expression of 29 protein-coding genes (eGenes) at FDR adjusted $P < 0.05$. We also identified 813 significant mQTL-DNAm associations consisting 158/273 mQTL SNPs (vs. 0.2% expected genome-wide[25], $P < 1 \times 10^{-300}$, hypergeometric test) associated with 778 DNAm sites (Supplementary Data 3, 4)."

**Co-occurring _cis_-eQTL and _cis_-mQTL.** We found an overlap between eQTL and mQTL at 23 SNPs (14.3% vs. 3.1% expected genome-wide[25], $P = 1.84 \times 10^{-11}$, hypergeometric test), forming a total of 197 eQTL-mQTL triplets (consisting 23 SNPs, 26 genes, and 152 DNAm sites) (Fig. 2a). The majority (88.5%) of *cis*-eQTLs were also *cis*-mQTLs (vs. 18.3% genome-wide[25], $P = 8.51 \times 10^{-19}$, hypergeometric test) (Fig. 2b). Among the 197 triplets, more SNPs were associated with gene expression and DNAm in opposite directions than in the same direction (68% vs. 32%), consistent with expectation that methylation near promoter and transcription start sites is related to inhibited transcription[26,27]. A significant inverse correlation was found between association coefficients of mQTL and eQTL in the placenta ($\rho = -0.22$, $P = 7.90 \times 10^{-157}$); the correlation strengthened when the analysis was restricted to 48 triplet subsets that included DNAm sites located within genes identical to the corresponding eGenes ($\rho = -0.58$, $P = 1.5 \times 10^{-5}$) (Fig. 2c, d). Further, we evaluated whether the co-occurring eQTL-mQTL triplets in the placenta have a similar relationship in blood using 23 triplets available in blood eQTL from GTEx[15] and middle-age blood mQTL from ARIES[28]. The inverse correlation between mQTL and eQTL coefficients in placenta remained ($\rho = -0.80$, $P = 6.61 \times 10^{-6}$) but no correlation was observed in blood ($\rho = 0.05$, $P = 0.82$) (Fig. 2e), suggesting that placenta may be the main organ in which these regulatory effects are targeted.

**Mediation analysis using the causal inference test.** For the 197 eQTL-mQTL triplets in which 23 co-occurring SNPs were associated with both placental gene expression and DNAm, we

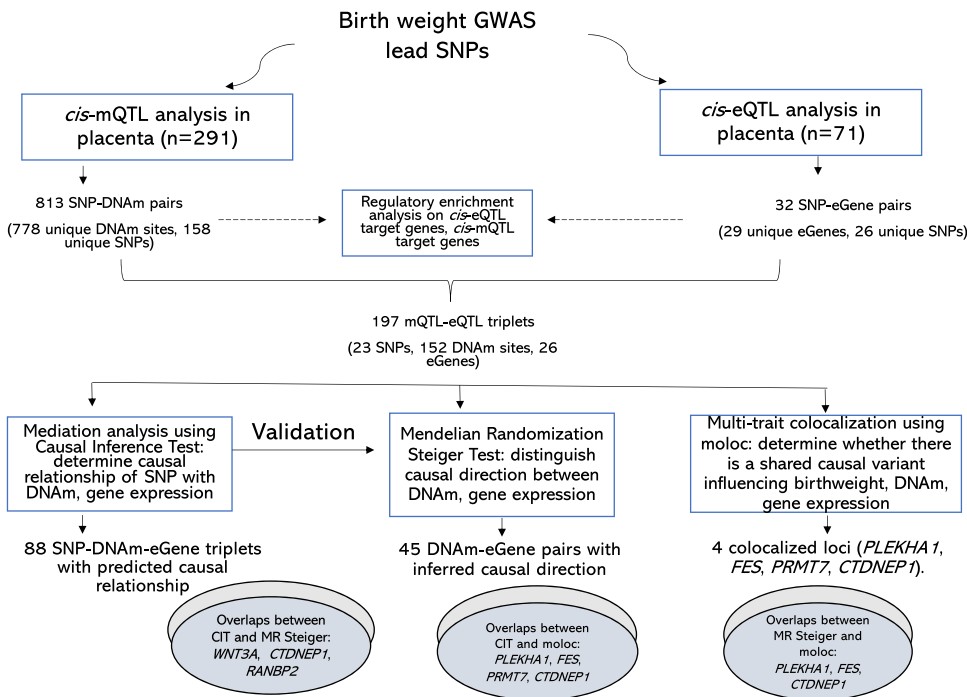

**Fig. 1 Overview of study workflow.** DNAm denotes DNA methylation, and eGene denotes an eQTL target gene in which gene expression in the placenta is regulated by a nearby genetic variant.

performed a causal inference test (CIT)[22] to distinguish among three scenarios in the causal relationship between DNAm and gene expression: (1) a birthweight GWAS SNP alters DNAm in the placenta, which in turn causally influences gene expression (SNP→DNAm→ gene expression, labeled as SME mediation). (2) a birthweight GWAS SNP alters gene expression in the placenta, which causally influences DNAm (SNP→gene expression→ DNAm, labeled as SEM mediation suggesting a passive role of DNAm on genetic regulation of gene expression[26], and (3) a birthweight GWAS SNP influences DNAm and gene expression in the placenta through independent pathways, but DNAm and gene expression have no causal relationship (SNP→gene expression or SNP→DNAm, denoted as independent SE/SM). CIT found evidence of a causal relationship in 44.7% (88/197) triplets consisting 15 SNPs, 17 eGenes, and 81 DNAm sites (Supplementary Data 5). Within the 88 predicted causal relationships, 84 (95.5%) were SME, predicting DNAm to causally influence gene expression and 4 (4.5%) were SEM, predicting gene expression to causally influence DNAm (Fig. 3a). There was a significant strong inverse dose-response correlation between the regression coefficients of mQTL and eQTL for triplets with predicted causal relationship (i.e., in the SEM and SME categories) ($\rho = -0.46$, $P = 8.13 \times 10^{-6}$) but not for triplets in the independent SE/SM category ($\rho = -0.15$, $P = 0.15$) (Fig. 3b, c), consistent with the causal inference findings. As expected, the distance between the DNAm site and the corresponding gene transcription start site was shorter for triplets with a predicted causal relationship (mean 134 bp) than independent or unclassified triplets (mean 183 bp and 224 bp, respectively).

**Mendelian Randomization analysis for validating direction of causality between DNAm and gene expression.** While the CIT has strength in distinguishing causal relationship from horizontal pleiotropy (independent pathways), it has limitations in distinguishing SME from SEM and might misclassify some causal relationships as independent[29]. To complement these limitations, we assessed the 197 eQTL-mQTL triplets using the Mendelian

Randomization (MR) Steiger directionality test[23]. The MR Steiger test provided evidence of significant inferred causal directions between DNAm and gene expression for 45 DNAm-eGene pairs consisting 11 SNPs, 37 DNAm sites, and 12 eGenes ($P < 0.05$; FDR $P < 0.05$ for 13 DNAm-eGene pairs, <0.1 for 36 DNAm-eGene pairs, <0.25 for all 45 DNAm-eGene pairs) (Supplementary Data 6). There was significant enrichment of DNAm-eGene pairs with causal directions (45/197 (22.8%) vs. 20.3% in a genome-wide dataset[30], $P = 1.17 \times 10^{-2}$, hypergeometric test). As expected, the majority (8/11; 72.7%) of SNPs with MR Steiger's evidence of inferred causal directions were among SNPs in CIT's predicted causal relationships (i.e., SME or SEM), and the remaining (3/11) were among SNPs in CIT's independent SE/SM category. Of the 45 DNAm-eGene pairs, 7 were from DNAm to gene expression (ME), and 38 were from gene expression to DNAm (EM). At the 7 ME DNAm-eGene pairs, the SNPs explained more variation in DNA methylation (ranging from 34.9% for cg01745370 to 74% for cg06106103) than in gene expression (ranging from 12.8% for *BTBD16* to 19.2% for *PLEKHA1*); therefore, the alternative explanation (i.e., reverse causal pathway from gene expression to methylation) is unlikely.

Consistent SME predictions were found by the CIT and MR Steiger test for three triplets (rs708122→cg02991924→*WNT3A* expression; rs222857→cg04106389 located in *DLG4*→*CTDNEP1* expression; and rs12104672→cg01745370 located in *GCC2* promoter region→ *RANBP2* expression; 3/45 vs. 88/197, $P = 2.96 \times 10^{-5}$, hypergeometric test). *WNT3A* is expressed in a few tissues only, its highest expression being in the placenta. Further diagnostic plots for inferring causal direction confirmed an SME relationship in rs708122→cg02991924→*WNT3A* expression (Fig. 4a–f). The three genes with causal directions triangulated by the two methods (i.e., *WNT3A*, *CTDNEP1*, and *RANBP2*) have been implicated by mice studies in placental development and vascularization, embryogenesis, and fetal growth[31–34].

**Multitrait colocalization.** The causal inference frameworks in CIT and MR Steiger tests cannot establish the possibility of

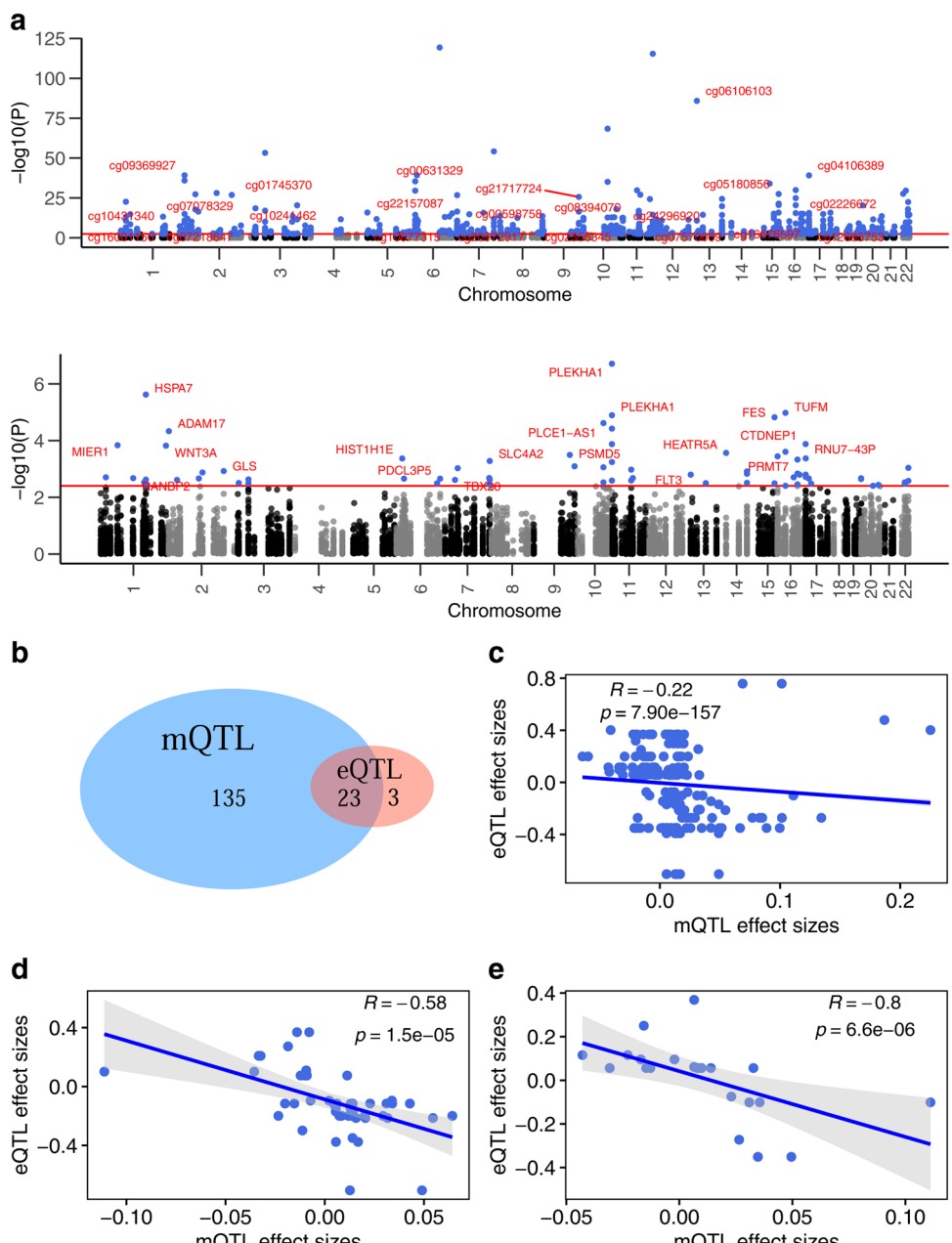

**Fig. 2 Birthweight-associated genetic variants with co-occurring *cis*-eQTL and *cis*-mQTL effect. a** mQTL in the upper panel and eQTL in the lower panel. The red horizontal lines represent the 5% FDR threshold. CpGs and genes with labels correspond to loci in which a genetic variant was associated with both DNA methylation and gene expression. For ease of readability, only one CpG and one gene was labeled per locus. **b** Venn Diagram showing the number of birthweight GWAS loci that were found to be eQTL and mQTL in the placenta. **c–e** *P* is based on a two-sided Spearman correlation test performed between placental mQTL and eQTL association coefficients for all 197 eQTL-mQTL triplets (**c**), for 48 triplets consisting DNA methylation sites within the eQTL target genes (**d**), and for 23 triplets that overlap with blood eQTL from GTEx and mQTL from ARIES database (**e**). **d, e** Error bands in gray represent standard error of the mean.

colocalized causal loci because the association of a variant with multiple traits may be due to distinct causal SNPs that are in linkage disequilibrium (LD). To determine whether there is a shared causal variant and identify functional genes underlying birthweight, DNA methylation and gene expression in the placenta, we performed a multitrait colocalization test using moloc[17]. A significant colocalization result implies that the alternative explanation that association signals are due to LD between the associated SNPs and separate causal variants for the traits is unlikely. We found evidence of genetic colocalization (posterior probability of association (PPA) ≥ 0.8) between

placental DNA methylation, gene expression, and birthweight at three loci and narrowly missed the cut-off at one locus: (1) *PLEKHA1* gene expression colocalized with DNAm sites located in *PLEKHA1* and *HTRA1* and birthweight (PPA = 0.85 and 0.88). (2) *FES* gene expression colocalized with nine DNAm sites in *FES* and birthweight (PPA = 0.82–0.91). (3) *PRMT7* gene expression colocalized with a DNAm site in *SMPD3* and birthweight (PPA = 0.8). (4) A fourth locus at chr17p13.1 showed evidence for colocalization of all three traits (*CTDNEP1* gene expression, DNAm sites in *DLG4* and *CLDN7*, and birthweight) that narrowly missed the cut-off (PPA = 0.77–0.78). At the chr17p13.1

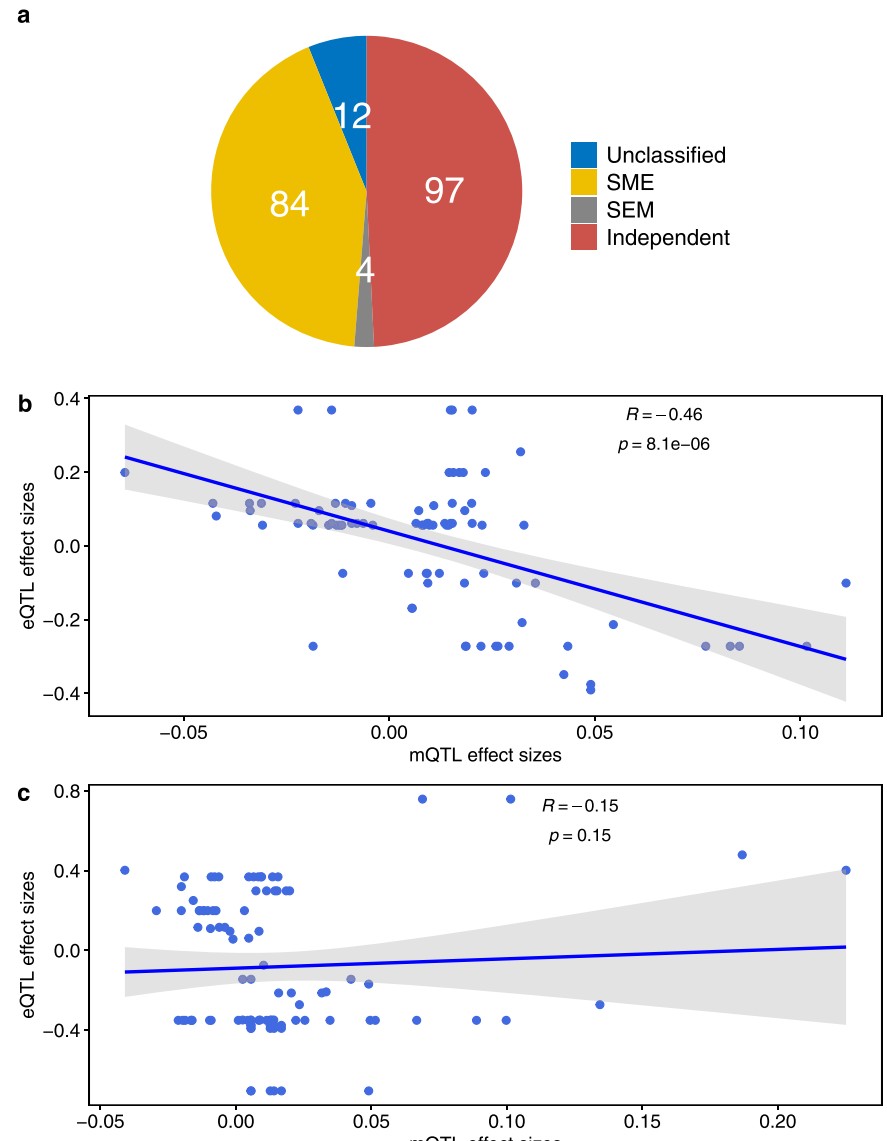

**Fig. 3 Causal inference test result. a** Pie-chart showing the number of eQTL-mQTL triplets predicted as "SEM" mediation representing SNP effect mediated via placental gene expression on nearby DNA methylation, "SME" mediation representing SNP effect mediated via placental DNA methylation on placental gene expression, "independent SE/SM" effect denoting SNP effect on placental DNA methylation and gene expression via independent pathways, and "unclassified" relationship. **b**, **c** Error bands in gray represent standard error of the mean and *P* is based on a two-sided Spearman correlation test. The dose-response correlation between eQTL and mQTL association coefficients was only significant for triplets with SEM or SME predicted causal relationship (**b**) but not with independent SE/SM relationship (**c**).

locus, there was a strong evidence of colocalization between birthweight and DNAm (PPA = 0.88–0.92), and between DNAm and *EIF5A* gene expression (PPA = 0.88–0.97) (Table 1).

For all four colocalized loci, the best-colocalized SNP identified by moloc was either the lead birthweight GWAS SNP (for *CTDNEP1*) or a nearby SNP in high LD ($r^2$ = 0.84–0.98 in 1000 Genomes European samples with the lead SNP) for *PLEKHA1*, *FES*, and *PRMT7*. For all four loci, the direction of association of the best-colocalized SNP with methylation was opposite to its association with gene expression. All four loci with evidence of colocalization overlapped with the triplets that had a predicted causal relationship by CIT and three loci overlapped with the triplets that had an inferred causal direction by MR Steiger test (M→E for *CTDNEP1*; E→M for *PLEKHA1* and *FES*). The colocalized genes are broadly expressed in several tissues, with the location of protein expression predicted as intracellular, and play key molecular functions including lipid-binding (*PLEKHA1*),

growth factor binding (*HTRA1*), chromatin regulator, kinase, and transferase (*PRMT7*, *FES*) (Supplementary Data 7). The colocalized DNAm sites are predominantly annotated to genomic regions with intron, promoter, and CpG shore features (Supplemental Fig. S1).

**Evaluation of colocalization in non-placental tissues**. We assessed whether the four loci with evidence of colocalization in the placenta also harbor shared causal variants for blood DNA methylation, gene expression, and birthweight. To do this, we applied moloc using summary statistics data for middle-age blood mQTL from ARIES[28], adult blood eQTL from GTEx[15], and GWAS of birthweight[6]. No significant (PPA ≥ 0.8) multitrait colocalization was found. There was suggestive evidence of colocalization between *PLEKHA1* gene expression in blood and birthweight (PPA = 0.76) (Supplementary Data 8). To expand our assessment to non-placenta tissues (with the caveat that

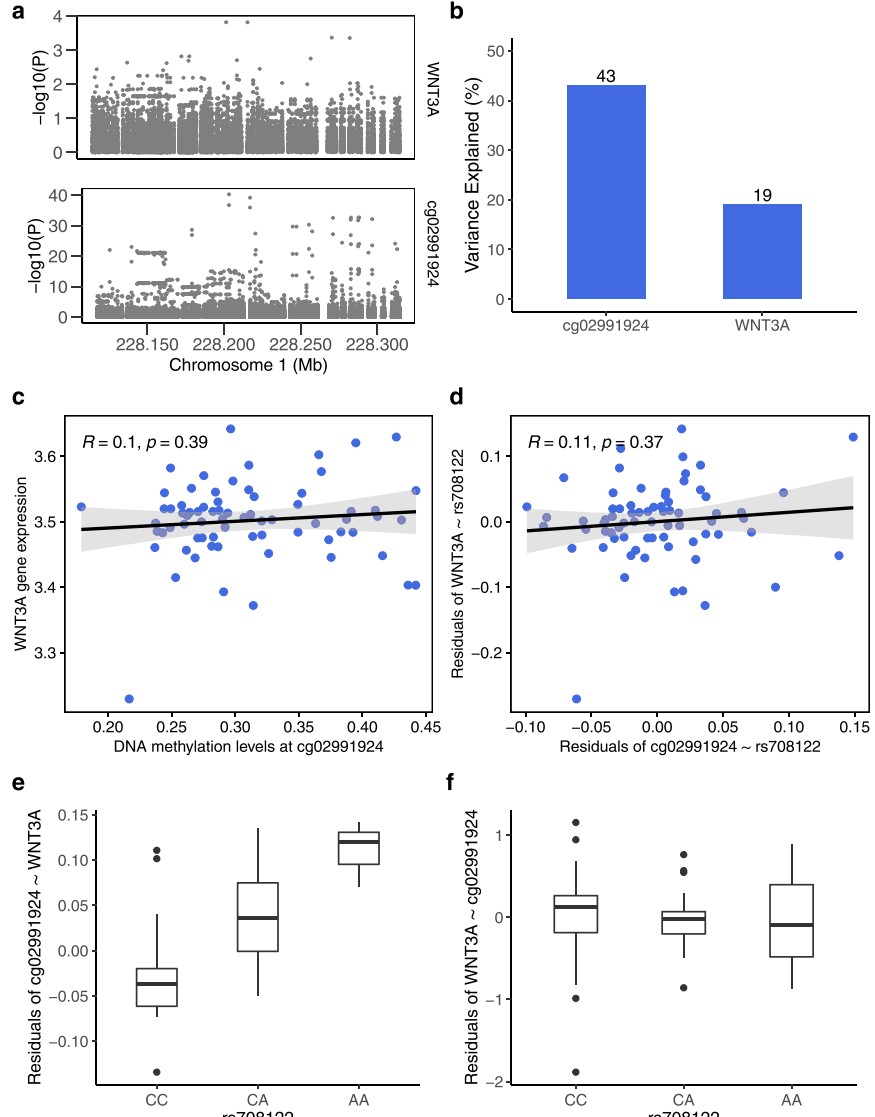

**Fig. 4 Direction of the relationship between birthweight-associated SNP, DNA methylation and *WNT3A* gene expression in placenta. a** Association between all SNPs 100 kb up- and downstream of the birthweight GWAS SNP rs708122 and *WNT3A* gene expression (upper panel) and DNA methylation at cg02991924 (lower panel). The strongest association was found for the index SNP or its nearest linkage disequilibrium (LD) proxies. **b** rs708122 accounts for higher variance in DNA methylation than in *WNT3A* expression based on MR Steiger test. **c, d** Error bands in gray represent standard error of the mean and *P* is based on a two-sided Spearman correlation test. DNA methylation and *WNT3A* expression were not correlated. **e, f** Box plots indicate median (middle line), 25th, 75th percentile (box) and 5th and 95th percentile (whiskers) as well as outliers (single points). *N* = 71 individuals. rs708122 was associated with DNA methylation after regressing out *WNT3A* expression (**e**) (Kruskal Wallis test $P = 3.06 \times 10^{-7}$) but was not associated with residuals of *WNT3A* after regressing out DNA methylation (**f**) (Kruskal Wallis test $P = 0.57$).

methylation data are not available), we tested colocalization for gene expression and the four birthweight loci in 48 GTEx tissues using coloc2[35]. Whereas no significant colocalization was found for *PLEKHA1* and *CTDNEP1*, evidence of colocalization was found for *PRMT7* and *FES* in some tissues (Supplementary Data 9).

**Evaluation of co-occurring *cis*-eQTL and *cis*-mQTLs in the placenta in an independent dataset.** Further assessment of the 197 co-occurring eQTL-mQTL pairs identified was done using placental data from the Rhode Island Child Health Study[24] (RICHS; *n* = 159). Data on 123 mQTLs and 22 eQTLs were available in RICHS. We found that 61% (75/123) mQTLs tested showed significant associations ($P < 0.05$) (vs. 0.2% genome wide[25], $P = 7.41 \times 10^{-169}$, hypergeometric test). Most of the

associated loci overlapped with the four colocalization target genes (*PLEKHA1*, *FES*, *CTDNEP1*, *PRMT7*) and the three genes showing consistent causal predictions by CIT and MR Steiger tests (*WNT3A*, *CTDNEP1*, *RANBP2*). 22.7% (5/22) eQTLs tested showed significant associations ($P < 0.05$) (vs. 0.04% genome-wide[25], $P = 5.40 \times 10^{-16}$, hypergeometric test). All eQTLs were also mQTLs in RICHS, mirroring the pattern we found in the NICHD Fetal Growth Studies dataset (Supplementary Data 10). The eQTLs for *PLEKHA1* and *RANBP2* are consistent with a previous report by Warrington et al.[6].

**Association between colocalized gene expression and birth-weight.** We assessed whether placental expression of the colocalized genes was associated with birthweight using linear regression adjusted for fetal sex and gestational age. *PLEKHA1*

**Table 1 Multitrait colocalization evidence for loci with causal-variant sharing between birthweight, placental DNAm, and gene expression.**

| GWAS lead SNP | SNP position (hg19) | eGene | DNAm site (nearest gene) | DNAm site position (hg19) | DNAm site relation to gene | Distance from GWAS SNP to eGene\|to DNAm site | N SNPs | PPA for G, EM | PPA for GM, E | PPA for GE, M | PPA for GEM |
|---|---|---|---|---|---|---|---|---|---|---|---|
| rs71486610 | Chr10:24134803 | PLEKHA1 | cg02556345 (PLEKHA1) | 124181965 | Body | Intronic \| 47.1 kb | 726 | $4.1\times10^{-2}$ | $3.7\times10^{-2}$ | $2.4\times10^{-2}$ | **0.88** |
| | | | cg06474225 (HTRA1) | 124228770 | Body | Intronic \| 93.9 kb | 686 | $4.1\times10^{-2}$ | $2.0\times10^{-2}$ | $3.2\times10^{-2}$ | **0.85** |
| | | | cg4366292 (DMBT1) | 124328787 | Body | Intronic \| 194.0 kb | 684 | $9.3\times10^{-5}$ | $7.8\times10^{-11}$ | **0.92** | $1.0\times10^{-8}$ |
| | | | cg17033087 (DMBT1) | 124329533 | Body | Intronic \| 194.7 kb | 682 | $1.2\times10^{-4}$ | $1.1\times10^{-5}$ | **0.90** | $1.3\times10^{-3}$ |
| | | | cg11976790 (DMBT1) | 124320064 | TSS | Intronic \| 185.3 kb | 686 | $1.7\times10^{-4}$ | $1.7\times10^{-12}$ | **0.88** | $2.7\times10^{-14}$ |
| | | | cg27015047 (DMBT1) | 124320805 | Body | Intronic \| 186.0 kb | 687 | $2.1\times10^{-4}$ | $1.4\times10^{-7}$ | | $1.3\times10^{-5}$ |
| rs4932373 | Chr15:91429287 | FES | cg25647583 (FES) | 91427784 | TSS | Intronic \| 2.1 kb | 507 | $4.6\times10^{-3}$ | $2.2\times10^{-2}$ | $4.9\times10^{-3}$ | **0.91** |
| | | | cg20992439* (FES) | 91429916 | Body | Intronic \| 0.6 kb | 510 | $2.4\times10^{-2}$ | $9.6\times10^{-3}$ | $1.7\times10^{-2}$ | **0.91** |
| | | | cg06330618 (FES) | 91428456 | Exon | Intronic \| 0.8 kb | 509 | $1.9\times10^{-2}$ | $1.9\times10^{-2}$ | $1.0\times10^{-2}$ | **0.90** |
| | | | cg09397246 (FES) | 91427361 | TSS | Intronic \| 1.9 kb | 507 | $6.2\times10^{-3}$ | $2.3\times10^{-2}$ | $1.7\times10^{-2}$ | **0.88** |
| | | | cg07718650 (FES) | 91434288 | Body | Intronic \| 5.0 kb | 527 | $2.1\times10^{-2}$ | $2.1\times10^{-2}$ | $1.6\times10^{-2}$ | **0.88** |
| | | | cg26405020 (FES) | 91427363 | TSS | Intronic \| 1.9 kb | 507 | $6.6\times10^{-3}$ | $2.3\times10^{-2}$ | $2.9\times10^{-2}$ | **0.86** |
| | | | cg08724371 (FES) | 91442212 | – | Intronic \| 12.9 kb | 544 | $8.4\times10^{-3}$ | $2.2\times10^{-2}$ | $3.4\times10^{-2}$ | **0.85** |
| | | | cg26899598* (FES) | 91429088 | Body | Intronic \| 0.2 kb | 509 | $1.1\times10^{-2}$ | $3.4\times10^{-2}$ | $2.9\times10^{-2}$ | **0.82** |
| | | | cg18661868* (FES) | 91427965 | 5'UTR | Intronic \| 1.3 kb | 508 | $1.6\times10^{-2}$ | $2.6\times10^{-2}$ | $3.1\times10^{-2}$ | **0.80** |
| rs1868158 | Chr16:68398924 | PRMT7 | cg02226672 (SMPD3) | 68398533 | Body | 6.4 kb \| 0.4 kb | 373 | $3.6\times10^{-2}$ | $3.0\times10^{-2}$ | $2.5\times10^{-2}$ | **0.80** |
| rs222857 | Chr17:7164563 | CTDNEP1** | cg17265693 (CLDN7) | 7166490 | TSS | 9.3 kb \| 1.9 kb | 434 | $4.5\times10^{-3}$ | $2.1\times10^{-2}$ | $2.6\times10^{-2}$ | **0.78** |
| | | | cg01697794 (DLG4) | 7117125 | Body | 9.3 kb \| 47.4 kb | 473 | $5.4\times10^{-3}$ | $2.0\times10^{-2}$ | $4.0\times10^{-3}$ | **0.77** |
| | | | cg19466160* (DLG4) | 7117160 | Body | 9.3 kb \| 47.4 kb | 473 | $5.1\times10^{-3}$ | $1.9\times10^{-2}$ | $4.6\times10^{-3}$ | **0.77** |
| | | EIF5A | cg17265693 (CLDN7) | 7166490 | Body | 45.8 kb \| 52.8 kb | 434 | $3.7\times10^{-7}$ | **0.92** | $2.8\times10^{-7}$ | $3.6\times10^{-5}$ |
| | | | cg01697794 (DLG4) | 7117125 | Body | 45.8 kb \| 58.1 kb | 473 | $6.1\times10^{-5}$ | **0.88** | $3.4\times10^{-7}$ | $3.2\times10^{-5}$ |
| | | | cg19466160 (DLG4) | 7117160 | Body | 45.8 kb \| 47.4 kb | 473 | $3.7\times10^{-6}$ | **0.88** | $3.9\times10^{-7}$ | $3.2\times10^{-5}$ |
| | | | cg17579446 (GPS2) | 7217367 | Body | 45.8 kb \| 52.8 kb | 396 | **0.97** | $4.8\times10^{-13}$ | $1.1\times10^{-7}$ | $1.9\times10^{-17}$ |
| | | | cg04514024 (NEURL4) | 7222668 | Body | 45.8 kb \| 58.1 kb | 403 | **0.88** | $1.9\times10^{-19}$ | $4.7\times10^{-7}$ | $3.3\times10^{-20}$ |

Chr chromosome.

Evidence of colocalization for a given scenario was based on the posterior probability of association (PPA) ≥ 0.8. For each locus indexed by the GWAS lead SNP, N SNPs denotes the number of SNPs within 1Mb from the index SNP. This table presents loci associated with all three traits and have evidence of colocalization in ≥2 of the traits birthweight, placental gene expression and placental DNA methylation (i.e, PPA ≥0.80 for any of the following four scenarios out of 15 Hypotheses tested by moloc): (1) PPA for G, EM corresponding H10: association for GWAS trait (G), eQTL trait (E) and mQTL trait (M), but different causal variants for {G}; and {E,M}. (2) PPA for GM, E corresponding H12: association for traits G, E and M, but different causal variants for {G,M} and {E}. (3) PPA for GE, M corresponding H11: association for traits G, E and M, but different causal variants for {G,E} and {M}. (4) PPA for GEM corresponding H14: SNP is associated with all 3 traits {G,E,M}. *CpG site has the promoter-associated regulatory feature. TSS denotes transcription start site. **PPA is very close to 0.8, or ≥0.8.

and *HTRA1* expression levels were significantly inversely associated with birthweight in the RICHS dataset and the association strengthened in meta-analyses of the results from RICHS and NICHD Fetal Growth Studies. *HTRA1* expression showed suggestive inverse association with placental weight in the NICHD Fetal Growth Studies ($P = 0.06$). Moreover, *PRMT7* and *CTDNEP1* expression levels showed directionally consistent inverse associations with birthweight in both datasets, although not reaching statistical significance (Supplementary Data 11). The inverse direction of association between these four genes and birthweight concurred with the eQTL findings such that the alleles with birthweight-lowering effect also had gene expression-increasing effect in the placenta. It is worth noting that gene expression levels are attributed to both genetic and environmental factors, and traits such as birthweight can in turn drive gene expression differences[36]. As a result, the relationship between *cis*-regulated gene expression and a trait may not be accurately mirrored by the relationship between measured gene expression and a trait. More accurate inference of the portion of gene expression, partitioned into genetic and non-genetic components, causally associated with birthweight warrants further investigation in larger well-characterized cohorts.

**Functional annotation, regulatory enrichment, and pathways.** Enrichment analysis of the 778 DNAm sites from our significant mQTL-DNAm pairs was performed using eFORGE[37]. We found significant enrichment in the placenta and depletion in the brain for DNase 1 hypersensitive sites (DHS), and enrichment in the placenta as well as other tissues for 15-state chromatin marks and H3 histone marks (Supplementary Data 12). The DNAm sites were enriched in CpG island shelve regions (FDR-adjusted $P < 0.05$) (Supplementary Data 13) and a striking majority (90%; 700/778) have been reported in epigenome-wide association studies (Supplementary Data 14).

Genes annotating the mQTL SNPs and their target DNAm sites ($n = 402$ genes) were significantly enriched (FDR-adjusted $P < 0.05$) in several ingenuity pathway analysis (IPA) canonical pathways related to immune response and endocrine signaling, in addition to metabolic and developmental pathways previously highlighted by GWAS of birthweight[6,7] (Supplementary Data 15). Consistent with this finding, gene set enrichment analysis (GSEA) found significant enrichment of upregulated hallmark gene sets associated with low oxygen levels (oxidative stress), immune response, adipocyte differentiation (adipogenesis), and development of skeletal muscle (myogenesis) and pancreatic beta cells (Supplementary Data 16). Significant enrichment was also observed for differential gene expression in GTEx tissues (Bonferroni-adjusted $P < 0.05$), with the strongest enrichment for upregulation of the gene set detected in subcutaneous adipose tissues relative to other GTEx tissues and the strongest enrichment for downregulation of the gene set detected in the brain relative to other GTEx tissues (Supplemental Fig. S2).

Birthweight-associated SNPs found to be eQTL in the placenta were significantly enriched for eQTL in 30 GTEx tissues; birthweight-associated SNPs that were mQTL in placenta were significantly enriched for mQTL in blood samples from the ARIES mQTL database[28] (Supplemental Fig. S3a–c).

**Discussion**

Using integrated multi-omics approaches involving the placenta, an organ essential in fetal development and implicated in later life health, we identified placental DNAm and gene expression regulatory targets for birthweight GWAS loci. We found 23 loci in which the birthweight GWAS lead SNP was concurrently associated with nearby gene expression and DNAm in the placenta.

For these co-regulated molecular traits, two complementary causal inference approaches found evidence of causal relationships between DNAm and gene expression. Multitrait colocalization prioritized four candidate causal genes (*PLEKHA1*, *FES*, *PRMT7*, and *CTDNEP1*) in which variations in placental DNAm, gene expression and birthweight are attributed to a single causal genetic variant. Together, these data suggest that the effect of the GWAS variants on birthweight is possibly mediated via their direct regulatory influence on epigenetic and transcriptomic changes in the placenta. Moreover, our finding that there is enrichment of immune and hormonal pathways among mQTL target genes suggests that the GWAS loci may influence birthweight via these processes, commensurate with recently reported roles of the placenta in immune regulation of maternal–fetal cross-talks[38–40], and hormonal regulation of fetoplacental growth[41–43].

We observed that SNPs associated with birthweight were more likely to be *cis*-mQTLs than *cis*-eQTLs in the placenta (158/273; 57.9% vs. 26/273; 9.5%) and only 14.6% of *cis*-mQTLs were *cis*-eQTLs. Similarly, in a previous study, less than one-third of DNAm changes associated with birthweight have been correlated with nearby gene expression in the placenta[44]. These observations may partly be explained by the diverse regulatory roles of DNAm beyond gene expression such as DNAse I accessibility, chromatin accessibility, and histone modification[27]. In contrast, the majority of *cis*-eQTLs were *cis*-mQTLs, which is much higher than the genome-wide overlaps between eQTLs and mQTLs in placenta[25], as well as other tissues[26,27,45]. Therefore, the birthweight GWAS loci that regulate placental gene expression may be tightly linked with epigenetic mechanisms in the placenta either as a mediator or a passive consequence.

Multitrait colocalization analysis identified four genes (*PLEKHA1*, *FES*, *PRMT7*, and *CTDNEP1*) in which a genetic variant associated with birthweight colocalizes with variations in DNA methylation and gene expression in the placenta. This evidence suggests that the association between the GWAS SNPs and birthweight at these loci might involve placental DNA methylation and transcription of nearby genes, and transcription may be a mechanism of effect consistent with causality. *PLEKHA1* is a GWAS locus for type 2 diabetes, adult height, and age-related macular degeneration[46–48]. Previously, evidence of colocalization has been found between *PLEKHA1* expression in the placenta and birthweight[49]. Besides replicating this finding, the present study provided additional evidence for colocalization with DNAm in *PLEKHA1* and *HTRA1*. *HTRA1* has the highest gene expression in human placenta and regulates cell growth by controlling the availability of insulin-like growth factors and TGF-beta signaling. Fetal and placental developmental abnormalities have been found in *htra1* knockout mice[50]. We observed that genetic variants associated with higher placental *HTRA1* expression have a birthweight-lowering effect. Similarly, placental expression of *htra1* is significantly increased in preeclampsia, a pregnancy complication associated with low birthweight[51]. It has been suggested that *htra1* overexpression in the placenta may disrupt microtubule organization, attenuated cell motility, and endothelial dysfunction[52,53], placental trophoblast invasion-related features critical in fetal growth.

Other salient exemplars of colocalized loci identified in our study include protein arginine N-methyltransferase 7 (*PRMT7*) and eukaryotic translation initiation factor 5A (*EIF5A1*). PRMT7 catalyzes arginine methylation, which is a post-translational modification that may play important physiological roles in DNA damage repair, transcriptional control, and establishment of the *Igf2/H19* imprinting control region methylation[54]. B cell-specific *prmt7*-knockout mice develop impaired immune response[55] and *Prmt7* was downregulated in the placenta of mothers fed high fat diet[56].

*EIF5A1* is crucial in the establishment of the placental–maternal circulation, placental development, and fetal–maternal immune tolerance[57,58].

Accumulating evidence shows considerable shared genetic overlap between fetal growth and cardiometabolic traits in later life[6,59–61]. Polygenic variants in the fetal genome associated with higher glucose and blood pressure have a birthweight-lowering effect. In contrast, polygenic variants in the maternal genome associated with higher glucose have a birthweight-raising effect possibly due to increased cross-placental transfer of glucose that induces insulin secretion and promotes fetal growth[59,62,63]. The birthweight-associated SNPs with colocalized effect on placental gene expression and methylation in our study have previously been associated with adult cardiometabolic phenotypes including blood pressure, type 2 diabetes, coronary artery disease, and height[6,59]. The colocalized genes we identified are broadly expressed in several tissues, but our evaluation in blood-derived eQTL from GTEx and mQTL from ARIES found no statistically significant evidence of multitrait colocalization. We also did not find significant eQTL-birthweight colocalization for *PLEKHA1* and *CTDNEP1* in any GTEx tissue. It is possible that the variants influence birthweight via the effector genes acting primarily in placenta, whereas the same variants may influence cardiometabolic traits in later life via other mechanisms. Multiple tissues with mQTL and eQTL data derived from identical samples will be critical to confirm this.

Our study has limitations. First, the maternal genome can indirectly influence placental molecular traits by altering the intrauterine environment. Some genetic loci with effects that vary by parent-of-origin have profound influence on placental development and function[64,65] and birthweight[59]. Understanding the genetic regulation of fetal growth is faced with the challenging task in delineating maternal genetic effects from fetal genetic effects, and in determining whether fetal effects depend on the variant's parental origin. We did not consider suggested classifications of the GWAS SNPs via the fetal and maternal genome because cluster assignment remains "unclassified" for several birthweight GWAS loci, and unambiguous assignment of these classifications is a work in progress. For example, some SNPs previously classified to act only via the maternal genome have been found to have additional effect via the fetal genome in a recent larger study with fetal, maternal, and paternal genotypes that improved the accuracy of genotype phasing[59]. Although it is reassuring that the colocalized loci we identified did not overlap with variants that have previously been found to influence birthweight via the maternal, but not fetal, genome[6,59], our study cannot distinguish direct fetal genetic effects from indirect maternal genetic effects. Second, given the heterogenous cellular architecture of the placenta[66], genetic effects that may be cell-type/state-specific need to be refined by future studies. Third, our analysis is likely underpowered to detect causality and colocalization at loci with weak eQTL or mQTL, warranting larger studies when placental datasets with multi-omics profiles become available. The size of our eQTL dataset is smaller than the mQTL dataset due to resource limitations, but is comparable to or larger than some tissue-specific datasets included in GTEx. The absence of significant difference in basic characteristics of the subset of study participants included in the eQTL analysis and those included in the mQTL analysis is reassuring that selection bias had minimal impact in our findings. Fourth, despite our study's strength that the candidate functional genes were identified through corroborating multi-omics evidence and replicated via mQTL in an independent cohort, definitive causal conclusions require future biological experiments. Last, the genetic variants associated with birthweight were discovered by studies involving largely European ancestry individuals; hence functional characterization of the genetic architecture of birthweight can be

improved when new loci are discovered via GWAS in non-European populations.

In conclusion, this study identified functional effector genes that underpin the genetic architecture of birthweight via placental epigenetic and transcriptomic mechanisms. Follow-up of the candidate causal genes identified in our study via experimental studies at the maternal–placental–fetal interface may unravel therapeutic targets to improve fetal growth outcomes and subsequent heath in later life.

## Methods

The NICHD Fetal Growth study protocol was approved by the institutional review boards of NICHD and each of the participating clinic sites, namely, Columbia University, New York; New York Hospital, Queens, New York; Christiana Care Health System, Delaware; Saint Peter's University Hospital, New Jersey; Medical University of South Carolina, South Carolina; University of Alabama, Alabama; Northwestern University, Illinois; Long Beach Memorial Medical Center, California; University of California, Irvine, California; Fountain Valley Hospital, California; Women and Infants Hospital of Rhode Island, Rhode Island; and Tufts University, Massachusetts.

**Dataset**. The discovery study included placenta samples obtained at delivery as part of the *Eunice Kennedy Shriver* National Institute of Child Health and Human Development (NICHD) Fetal Growth Studies—Singletons in which 2802 pregnant women in the United States were recruited between 8 and 13 gestational weeks and followed through delivery. Women with low risk for adverse pregnancy complications were recruited from four race/ethnic groups (i.e., non-Hispanic White, non-Hispanic Black, Hispanic, and Asian or Pacific Islander) from July 2009 to January 2013. To be enrolled, women had to have no past adverse pregnancy outcomes and no major pre-existing medical conditions including autoimmune diseases, chronic hypertension, diabetes, chronic renal disease, cancer, HIV/AIDS, or psychiatric disorders. Gestational age was determined using the date of the last menstrual period and confirmed by ultrasound between 8 weeks to 13 weeks and 6 days of gestation[20,21]. Written informed consent was obtained from all study participants.

Placental biopsies measuring 0.5 cm × 0.5 cm × 0.5 cm were taken from the fetal side ($n = 312$) within 1 h of delivery and samples were placed in RNAlater and frozen $-80$ °F for molecular analysis[25,67]. DNA extracted from the placental biopsies was genotyped using HumanOmni2.5 Beadchip (Illumina Inc., San Diego, CA) and DNAm was profiled on the 312 samples using Illumina's Infinium Human Methylation450 Beadchip (Illumina Inc., San Diego, CA). We removed probes with mean detection $P \geq 0.05$, cross-reactive, non-autosomal, and CpG sites located within 20 base pair from known SNPs. Samples showing discrepancies between phenotypic sex and genotypic sex, outliers from the distribution of the samples' genetic clusters, and with a mismatching sample identifier were excluded. RNA from a subset of the placental samples ($n = 80$) was extracted using TRIZOL reagent (Invitrogen, MA), and RNA sequencing was performed using the Illumina HiSeq2000 system with 100 bp paired-end reads. The reads were mapped to the human reference genome (NCBI/build 37.2) using Tophat version 2.0.4. On the genotype dataset, we removed SNPs with >5% missing values, minor allele frequency <0.5%, and not in Hardy–Weinberg equilibrium ($P < 10^{-4}$). SNP genotypes were imputed using the Michigan Imputation Server using the 1000 Genomes Phase 3 genotype reference [68], and filters were applied to remove insertion–deletions, SNPs with minor allele frequency <0.5% and SNPs with imputation dosage $r^2 < 0.3$[8,12,25,67]. The present analysis included samples that passed quality control, i.e., 291 samples with fetal genotype and DNAm data for mQTL analysis, of which 71 samples also had RNA-seq data for eQTL analysis.

**Identification of placental *cis*-eQTL for GWAS variants associated with birthweight**. SNPs known to be associated with birthweight at $P < 5 \times 10^{-8}$ (i.e., 286 SNPs consisting 190 lead SNPs with $P < 6.6 \times 10^{-9}$ and 96 SNPs with $6.6 \times 10^{-9} < P < 5 \times 10^{-8}$) were extracted from a GWAS of birthweight by Warrington et al.[6]. Out of the 286 SNPs, 273 (consisting 180 of the 190 lead SNPs and 93 of the 96 SNPs) were available in our dataset (Supplementary Table S2). A total of 7901 mRNA transcripts within 1 Mb from the 273 SNPs were included in *cis*-eQTL analysis. *cis*-eQTL analysis was performed with QTLtools software[69] using linear regression under an additive genetic model for each SNP, adjusted for fetal sex, self-reported race/ethnicity, top ten genotype principal components (PCs) to account for population structure, and top three RNA-seq PCs. The quantile-quantile (QQ) plot of $P$-values resulting from the *cis*-eQTL mapping of the 273 SNPs and all 7901 genes within 1-MB distance from the SNPs showed an absence of inflation ($\lambda = 0.93$) (Supplemental Fig. S4). Statistical significance was considered based on a false discovery rate (FDR) of 0.05.

**Identification of placental *cis*-mQTL for GWAS variants associated with birthweight**. A total of 104,053 DNAm sites within 1 Mb from the 273 SNPs were included in *cis*-mQTL analysis. *cis*-mQTL analysis was performed using linear

regression under an additive genetic model for each SNP, adjusted for fetal sex, ethnicity, methylation plate, top ten genotype PCs, and top three methylation PCs as implemented in QTLtools[69]. Methylation PCs were included in the model as covariates to remove the hidden batch effects and other potential confounders in the DNAm data. The QQ plot of $P$-values resulting from the mQTL mapping of the 273 SNPs and all 104,053 CpG DNAm sites within 1-MB distance from the SNPs showed an absence of inflation ($\lambda = 0.99$) (Supplemental Fig. S5). Statistical significance was considered based on FDR of 0.05.

**Causal inference test**. For birthweight GWAS SNPs that were found to be both eQTL and mQTL in placenta, mediation analysis was performed to determine the direction of causal relationship of the SNP with DNAm and gene expression using the causal inference test (CIT v2.2)[22]. CIT is a mediation-based approach that performs a series of conditional regression tests to predict a causal relationship from SNP to exposure to outcome. The CIT outputs $P$-values of a causal model (SNP → DNAm → gene expression; pCausalCIT) and a reverse causal model (SNP → gene expression → DNAm; pRevCausalCIT). Predicted causal direction was assigned as SME when pCausalCIT <0.05 and pRevCausalCIT ≥0.05, and as SEM when pCausalCIT ≥0.05 and pRevCausalCIT <0.05. Triplets with pCausalCIT ≥0.05 and pRevCausalCIT ≥0.05 were considered independent and triplets with pCausalCIT <0.05 and pRevCausalCIT <0.05 were considered unclassified.

**Mendelian Randomization (MR) Steiger test to validate the direction of causality between DNA methylation and gene expression**. To distinguish the causal direction between DNAm and gene expression for mQTL-eQTL triplets in which a co-occurring SNP was associated with DNAm and gene expression, we performed MR Steiger directionality test[23]. The MR Steiger test distinguishes the causal direction between exposure and outcome by testing whether the variance in the outcome explained by the instrumenting SNP is less than the variance in the exposure explained by the instrumenting SNP. Summary statistics from placental *cis*-mQTL and *cis*-eQTL were used as input for the MR Steiger test implemented in the TwoSampleMR package[29].

**Multitrait colocalization**. We used a Bayesian multiple-trait colocalization analysis implemented in the R package moloc[17] to investigate whether the same causal variant influenced birthweight, placental methylation at a nearby DNAm site and gene expression at a nearby gene. Input data for moloc were GWAS summary data for "own birthweight" from the recent GWAS of birthweight[6] (i.e., regression coefficients, their variances, and SNP minor allele frequencies in Europeans) for 23 SNPs with co-occurring *cis*-eQTL and *cis*-mQTL effects, placental eQTL data (spanning all genes in which the transcription start site is within 1 MB of a given SNP) and placental mQTL data (spanning all CpG DNAm sites within 1 MB of a given SNP) derived from our analysis. All 23 loci were analyzed because they had 50 or more SNPs with minor allele frequency >5% within their genomic region in common between all three datasets as recommended[17]. Given three traits were evaluated in our analysis, moloc computed 15 possible configurations of the association between genetic variation and traits. For each such locus, colocalization analysis was performed using default parameters and a posterior probability of association ≥80% was considered evidence of colocalization as recommended[17]. Loci with evidence of colocalization suggest presence of a single causal variant that influences DNA methylation, gene expression, and birthweight.

Next, we assessed whether loci found to have evidence of genetic colocalization in our analysis also harbor shared causal variants for blood DNAm, blood gene expression, and birthweight. To do this, we applied moloc using summary data of blood mQTL at middle-age from the ARIES mQTL database[28], blood eQTL from the GTEx Portal[15], and GWAS of birthweight[6].

**Assessment of the eQTL-mQTL pairs in an independent dataset**. The 197 eQTL-mQTL pairs identified using the NICHD Fetal Growth Studies dataset were additionally assessed in the Rhode Island Child Health Study (RICHS) dataset. RICHS recruited mother and infant pairs from March 2009 to May 2013 following delivery at the Women and Infants Hospital of Rhode Island. RICHS selected infants born small for gestational age, large for gestational age and controls born appropriate for gestational age matched on sex, gestational age (±3 days), and maternal age (±2 years)[24]. The study protocol was approved by the Institutional Review Boards of Brown University and Women and Infants Hospital of Rhode Island. Placental RNA-seq data from a subset of samples ($n = 200$) were obtained using the Illumina Hi-Seq 2500 platform, aligned to the human reference genome, and RNA transcript abundance was quantified using Salmon[70]. About 20 million single-end RNA-seq reads were generated on each sample[71]. Placental DNAm data ($n = 220$) were obtained using the Infinium MethylationEPIC array (Illumina), preprocessed, and normalized with the minfi R package[72]. Genotype data were available for 159 infants obtained using the Illumina MegaEX array and imputed using the Haplotype Reference Consortium reference panel[73]. eQTL and mQTL analyses were performed on the 159 samples with genotype, RNA-seq, and DNAm data.

Of the 23 SNPs associated with DNAm and gene expression in the 197 eQTL-mQTL pairs in our NICHD Fetal Growth Studies cohort, 22 SNPs were present in RICHS dataset (including rs77553582 used in place of rs7790713 which was

missing in the dataset [$r^2 = 1$ in 1000 Genomes EUR]). Of the 26 eGenes for *cis*-eQTL assessment, 20 eGenes were present in RICHS data after quality control and were *cis* with 18 SNPs. Of the 152 DNAm sites for *cis*-mQTL assessment, 122 DNAm sites were present in RICHS data after quality control, and 121 were *cis* with 22 SNPs. A total of 22 *cis*-eQTL tests were performed between the 18 SNPs and 20 eGenes, and a total of 123 *cis*-mQTL tests were performed between the 22 SNPs and 121 DNAm sites ($n = 159$) using QTLtools software[69]. Analyses were adjusted for sex, self-reported ethnicity, methylation or RNA-seq batch, top ten genotype PCs, and top three expression PCs (for eQTL) or methylation PCs (for mQTL).

**Functional annotation and regulatory enrichment**. For the 778 unique DNAm sites from the significant mQTL-DNAm pairs, we assessed enrichment and depletion of overlap with tissue-specific or cell-type-specific regulatory features including DNase 1 hypersensitive sites (DHS), all 15-state chromatin marks, and all five H3 histone marks (i.e., H3K27me3, H3K4me1, H3K4me3, H3K36me3, H3K9me3) using eFORGE v2.0[37]. The set of 778 DNAm sites was entered as input in eFORGE and tested separately for enrichment and depletion of overlap with each of the three putative functional elements (i.e., DHS, all 15-state chromatin marks, and all five H3 marks) compared to the respective data from consolidated ROADMAP epigenomics. The enrichment analysis was performed in one thousand matched background sets, with each background set consisting 778 DNAm sites matched for gene and CpG island annotation. Regulatory regions were also tested for enrichment in relation to their locations from CpG island genomic regions (island, shelves, shores, and open sea) based on functional information from the Encyclopedia of DNA Elements (ENCODE)[74] and Roadmap Epigenomics Project[75]. Enrichment or depletion was considered statistically significant at FDR < 0.05.

**Pathway analysis**. Pathway analysis was performed using the Ingenuity Pathway Analysis tool (IPA, QIAGEN, Redwood City, CA, USA, www.qiagen.com/ingenuity). We used IPA for two sets of genes: (1) genes near the SNPs and CpG sites in the significant mQTL-DNAm pairs (a set of 402 unique genes), and (2) genes near the SNPs and the target genes in the significant eQTL-eGene pairs (a set of 50 unique genes). Enrichment in canonical pathways was assessed using the right-tailed Fisher's exact test. Pathways with three or more molecules enriched at FDR < 0.05 were considered statistically significant. Furthermore, the abovementioned two sets of genes were annotated in biological context using the GENE2FUNC option using FUMA, a web-based platform that facilitates functional annotation of GWAS results[76]. FUMA provides hypergeometric tests of enrichment of a list of genes in 54 GTEx tissue-specific gene expression sets (GTEx v 8). Gene set enrichment analysis was performed using FUMA to test whether the input genes were enriched in hallmark gene sets, which are biological states displaying coordinated expression as defined by the Molecular Signatures Database (MsigDB v7.2)[77].

**Reporting summary**. Further information on research design is available in the Nature Research Reporting Summary linked to this article.

## Data availability

The genotypes, DNA methylation, and gene expression data used in this study from the NICHD Fetal Growth Studies have been deposited in the dbGaP database under accession code phs001717.v1.p1. The genotypes and gene expression data from the Rhode Island Child Health Study (RICHS) have been deposited in the dbGaP database under accession code phs001586.v1.p1. GWAS summary statistics for birthweight are available via the EGG website (https://egg-consortium.org/). The 1000 Genomes Reference Panel datasets are available at https://www.internationalgenome.org/category/reference/. The human genome reference is made accessible by the Genome Reference Consortium at https://www.ncbi.nlm.nih.gov/assembly/GCF_000001405.13/. GTEx eQTL data are accessible via the GTEx Portal at https://gtexportal.org/home/eqtlDashboardPage. ARIES mQTL database is available for download at http://www.mqtldb.org/. Genotype imputation platform as well as the Haplotype Reference Consortium reference panel can be accessed via the Michigan Imputation Server at https://imputationserver.readthedocs.io/en/latest/. The catalog of epigenome-wide association studies can be accessed at http://ewascatalog.org/. Regulatory feature annotations of CpG sites are accessible from ENCODE and Roadmap Epigenomics projects at https://www.encodeproject.org/ and http://www.roadmapepigenomics.org/data/. Functional mapping of SNPs and genes via FUMA is accessible at https://fuma.ctglab.nl/.

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

## Acknowledgements

This work was supported by the Intramural Research Program of the *Eunice Kennedy Shriver* National Institute of Child Health and Human Development (NICHD), National Institutes of Health (NIH) including American Recovery and Reinvestment Act funding via contract numbers HHSN275200800013C; HHSN275200800002I; HHSN27500006; HHSN275200800003IC; HHSN275200800014C; HHSN275200800012C; HHSN275200800028C; HHSN275201000009C and HHSN27500008. Additional support was obtained from the NIH Office of the Director, the National Institute on Minority Health and Health Disparities (NIMHD) and the National Institute of Diabetes and Digestive and Kidney Diseases (NIDDK). RICHS is partially supported by NIH-NIEHS R01ES022223 (CJM, JC and KH), NIH-NIEHS R01ES022223-03S1 (CJM), and NIH-NIEHS R24ES028507 (CJM). CL is funded through NIH-NICHD R00HD097286. The authors acknowledge the research teams at all participating clinical centers for the NICHD Fetal Growth Studies, including Christina Care Health Systems, Columbia University, Fountain Valley Hospital, California, Long Beach Memorial Medical Center, New York Hospital, Queens, Northwestern University, University of Alabama at Birmingham, University of California, Irvine, Medical University of South Carolina, Saint Peters University Hospital, Tufts University, and Women and Infants Hospital of Rhode Island. The authors also acknowledge C-TASC and The EMMES Corporations in providing data and imaging support. Genotyping was performed in the Department of Laboratory Medicine and Pathology, University of Minnesota. This work utilized the computational resources of the NIH HPC Biowulf cluster (http://hpc.nih.gov). The authors are also thankful to the RICHS study participants for their participation, and the study staff at Women and Infants Hospital for their dedication to the project. Data on birthweight GWAS summary statistics have been contributed by the EGG Consortium using the UK Biobank Resource and has been downloaded from www.egg-consortium.org.

## Author contributions

F.T.-A. conceived and designed this study and wrote the draft manuscript. X.Z., S.C., and M.O. performed statistical analyses and visualizations. C.L. analyzed the replication dataset M.T. reviewed functional annotations. T.W. contributed to data quality control. K.H. and J.C. contributed the data. C.J.M. and R.W. contributed data and samples. F.T.-A., X.Z., S.C., M.O., C.L., K.H., J.C., M.T., C.J.M., T.W., and R.W. interpreted the results, reviewed the draft manuscript, provided critical intellectual content, and approved the final manuscript.

## Funding

## Competing interests

The authors declare no competing interests.
