## [Peer Review File · Nature Communications]

Placental multi-omics integration identifies candidate functional genes for birthweightReviewers' Comments:

Reviewer #1:

Remarks to the Author:

1. The authors used the birth weight GWAS of Warrington et al 2019. Supplementary Tables 5a and b of the Warrington et al paper reports placental eQTL genes from the GWAS candidate also using the Rhode Island Child Health Study. Have they looked to see whether there is overlap between their study and the 2019 study?
2. Multi-trait co-localization identified a number of key genes. However, I did not see an analysis of whether the expression levels of these genes in placenta were associated with birth weight, and this should have been possible using the RICH study data. If there is an association, they should present the data as it considerably strengthens their case. However, if there is no association, can they explain why?
3. The authors studied 286 SNPs from the Warrington et al 2019 study. However, the study itself reports that there were 209 lead SNPs. Moreover, the 209 consisted of SNPs with maternal only effects, fetal only effects, and both maternal and fetal effects, which was further subdivided by whether the maternal and fetal effects were in the same or opposite directions. Can they explain the larger number of SNPs than the original paper? Also, why include SNPs which are maternal only as these would be unlikely to have direct effects of placental gene methylation or gene expression?
4. Birth weight is jointly determined by relative fetal growth and by gestational duration. Warrington et al 2019 et al adjusted analysis of birth weight for fetal sex and gestational duration. I searched the MS for the word "gestational" and could not find any reference to adjustment for gestational duration. Could they please clarify if they simply analyzed crude birth weight? If so, this is a significant weakness. A 2500g birth weight would be normal at 37 weeks but small for gestational age at 41 weeks.
5. Lines 99-103. There are a series of proportions indicating overlap. But some overlap would be expected by the play of chance. It would be useful to know in each of the cases of overlap what the P value was for the observed versus the expected overlap based on the null hypothesis.
6. Lines 140-142. Was the $P < 0.05$ corrected for a 5% FDR? Was the finding of 45 DNAm-eGene pairs greater than would have been predicted by the play of chance? How many pairs were $P < 0.05$ with a 5% FDR?
7. Lines 151-154. Consistent predictions were observed using CIT and MR Steiger for three triplets. Is this more than would have been expected by the play of chance?
8. Lines 194-202. External validation is crucial in demonstrating that the findings are correct and generalizable. Was the degree of overlap in the findings of the main study and the RICH study more than would be consistent with the play of chance? This should be spelled out in terms of the main findings, the observed overlap, the expected overlap based on the null hypothesis, and the P value for comparing observed versus expected based on the null hypothesis.

Reviewer #2:

Remarks to the Author:

This manuscript performed a multi-omics "post-GWAS" approach to mining the intermediate markers of DNA methylation and gene expression to provide mechanistic insight into genetic SNPs associated with birthweight. Placenta tissue from a total of 71 births (+159 in replication cohort) for which imputed SNPs, Infinium EPIC array, and RNA-seq data (performed in previous studies) were all available. They started with 273 SNPs previously associated with birthweight and limited the analysis of mQTLs and eQTLs to 1 MB from each SNP, resulting in 197 mQTL-eQTL "triplets" mapping to 26 genes. They then performed 2 statistical tests of causality of directionality and an additional multi-trait colocalization test on the triplets. The following gene loci exhibited evidence for mQTLs with methylation causally influencing expression in placenta: WNT3A, CTDNEP1, and RANBP2. The following loci showed significant multi-trait colocalization: PLEKHA1, FES, CTDNEP1, and PRMT7. These results provide important functional insight into how birthweight GWAS SNPs may functionally interact with

epigenetic mechanisms to regulate gene expression in placenta. Overall, the study appears to be well performed and described. However, there are a number of places where the approaches and populations in the study could be better described.

1. The authors admit that a limitation to their study was the lower number of placental samples for which RNA-seq was performed compared to EPIC. But the reason for this was not explained but important to include. For instance, did it have to do with RNA quality, budget, lack of overlap in existing studies, or some other type of selection? The concern would be to try to reduce the impacts of selection bias in the relatively small sample size of 71 participants for the discovery group.
2. While the study was based on multi-omics data that was individually published previously, the authors should provide information about study participants specifically used in this study, including variables such as age, gestational age, parity, maternal BMI, child sex, race/ethnicity, etc. This should be provided for both the discovery and replication cohorts.
3. The wording describing results was sometimes imprecise, particularly in the sections describing enrichments (lines 203-222). The terms "upregulated" and "downregulated" are meaningless without context. Using phrases such as "associated with birthweight" and other more precise language in this section would help improve clarity. Other places in the text could also use reminders to the reader that the associations were with birthweight.
4. The "author contribution" section only describes the contributions of 3 out of 9 authors. While likely an inadvertent oversight, this is concerning if authors are included who did not contribute in a meaningful way.

Reviewer #3:

Remarks to the Author:

The goal of this study is to provide functional mechanistic insights linking genetic variation to birthweight (BW). GWAS variants associated with BW were linked to methylation and gene expression data specific to the placenta. Findings suggest a causal role for a specific set of genes, previously linked to cardiometabolic, immune and hormonal regulation. Overall, the novelty of the approach, particularly the application to the placenta, makes this an interesting study that that will form the basis of several future investigations. The main problem with the data generated is the lack of any direct evidence that the identified associations are (i) specific in any way to the placenta and (ii) that they impact placental functioning in a manner that might explain variation in BW.

Strengths

- Very interesting study design and an important hypothesis that has largely been ignored to date. The three-step approach to build evidence linking specific BW associated SNPs to placental expression and methylation is really nice
- Reasonably large sample size of 291 samples (with ethnic diversity) with both SNP and methylation data, some with gene expression data. Overall, the sample is adequate to reveal some novel findings, but is far from conclusive
- Use of independent dataset provides further support for the robust nature of the findings is good
- Several of the study limitations are highlighted

Limitations

- Recent studies suggest a contribution of both indirect maternal and direct fetal genetic effects on BW. It is now clear that performing GWAS solely on fetal genetics precludes full resolution of the origin of the identified genetic effects. In other words, maternal genetic factors are also likely very important in regulation of placental structure and function, and therefore, BW
- The most recent examination of the genetic influences on BW identified a range of biological pathways that contribute to this phenotype including variants influencing blood pressure, glycemic

traits and height (amongst others). Few of these would have any link to placental function but should not be disregarded as important determinants of BW

- More information is needed on the starting 286 BW associated SNPs. Warrington et al, identified 190 independent SNP associations derived from 146 offspring own BW and 72 maternal effect SNPs. How does this impact the interpretation of the current study where only offspring genotype was considered?

- 273 SNPs were available in the placental dataset but there is no information given on how many of the variable genotypes were actually present and at what frequency. This should be included as a table

- If not already included, it is important that the specific SNP, CpG and genes of interest be included in the current study, particularly the distance from sites of interest to the genes linked to them.

- GTEx data from non-placental tissues indicates a likely large component of these tissues to the observed effects on BW. This needs to be discussed more

- Little data on the localisation of expression of genes of interest within the placenta is shown.

- Very little information is given on the relative localisation of specific CpGs, SNPs or gene elements (promoters, exons etc), particularly those listed in Table 1.

Minor points

- Although true a decade ago, it is disputable whether the placenta remains understudied in terms of genomics. Despite this, its exclusion from important international initiative such as GTEx has been unfortunate.

Response to reviewers' comments

Reviewer #1 (Remarks to the Author):

1. The authors used the birth weight GWAS of Warrington et al 2019. Supplementary Tables 5a and b of the Warrington et al paper reports placental eQTL genes from the GWAS candidate also using the Rhode Island Child Health Study. Have they looked to see whether there is overlap between their study and the 2019 study?

Thank you for your careful review and detailed feedback. We have examined overlaps with Warrington's report. For *PLEKHA1*, we identified co-occurring eQTLs and mQTLs in both the NICHD Fetal Growth Study and the Rhode Island Child Health Study datasets; similarly, an eQTL has been reported by Warrington et al 2019. For *RANBP2*, we identified co-occurring eQTLs and mQTLs in the NICHD Fetal Growth Study dataset and an eQTL in the Rhode Island Child Health Study dataset; similarly, an eQTL has been reported by Warrington et al 2019. We did not see overlaps in other eQTL and mQTL genes, most likely because the Rhode Island Child Health Study was included at the replication step for selected loci that were found to be both eQTL and mQTL in the NICHD Fetal Growth Study discovery dataset as well as being a GWAS locus, whereas Warrington et al assessed eQTL across the GWAS loci associated with birthweight. As mentioned in the Discussion section (line 298-299), *PLEKHA1* has also been found to be a colocalization signal in a prior work. We have now expanded the results section adding the following statement:

Line 225-226: "The eQTLs for *PLEKHA1* and *RANBP2* are consistent with a previous report by Warrington et al⁶"

2. Multi-trait co-localization identified a number of key genes. However, I did not see an analysis of whether the expression levels of these genes in placenta were associated with birth weight, and this should have been possible using the RICH study data. If there is an association, they should present the data as it considerably strengthens their case. However, if there is no association, can they explain why?

As suggested, we have now tested this association in both RICHS and NICHD Fetal Growth Studies datasets, and the results are presented in Table S11 and the following statement has been included.

Line 227-243: Association between colocalized gene expression and birthweight. We assessed whether placental expression of the colocalized genes was associated with birthweight using linear regression adjusted for fetal sex and gestational age. *PLEKHA1* and *HTRA1* expression levels were significantly inversely associated with birthweight in the RICHS dataset and the association strengthened in meta-analyses of the results from RICHS and NICHD Fetal Growth Studies. *HTRA1* expression showed suggestive inverse association with placental weight in the NICHD Fetal Growth Studies ($P=0.06$). Moreover, *PRMT7* and *CTDNEP1* expression levels showed directionally consistent inverse associations with birthweight in both datasets, although not reaching statistical

significance (**Table S11**). The inverse direction of association between these four genes and birthweight concurred with the eQTL findings such that the alleles with birthweight-lowering effect also had gene expression-increasing effect in placenta. It is worth noting that gene expression levels are attributed to both genetic and environmental factors, and traits such as birthweight can in turn drive gene expression differences³⁶. As a result, the relationship between *cis*-regulated gene expression and a trait may not be accurately mirrored by the relationship between measured gene expression and a trait. More accurate inference of the portion of gene expression, partitioned into genetic and non-genetic components, causally associated with birthweight warrants further investigation in larger well-characterized cohorts.

Table S11. Association of placental expression of genes identified in multi-trait colocalization with birthweight and placental weight

3. The authors studied 286 SNPs from the Warrington et al 2019 study. However, the study itself reports that there were 209 lead SNPs. Moreover, the 209 consisted of SNPs with maternal only effects, fetal only effects, and both maternal and fetal effects, which was further subdivided by whether the maternal and fetal effects were in the same or opposite directions. Can they explain the larger number of SNPs than the original paper? Also, why include SNPs which are maternal only as these would be unlikely to have direct effects of placental gene methylation or gene expression?

We have now clarified in the Results and Methods sections that the 286 SNPs consist of all SNPs in Warrington et al 2019 study passing the GWAS significance threshold of $P < 5 \times 10^{-8}$ (i.e., 190 lead SNPs (from a total of 209 SNPs) with $P < 6.6 \times 10^{-9}$ and 96 SNPs with $6.6 \times 10^{-9} < P < 5 \times 10^{-8}$). [Note that in a look-up of the 96 loci in a larger study published while our manuscript was under review (*Nat Genet* 53, 1135-1142 (2021)), we observed that more than 1/3rd were found to be genome-wide significant at class-based significance thresholds, strengthening our decision to include all loci passing the GWAS threshold]. We have now added the list of analyzed SNPs in a Table S2 and clarified the Results and Methods sections as follows:

Line 91-93: Out of 286 single nucleotide polymorphisms (SNPs) found to be associated with birthweight at $P < 5 \times 10^{-8}$ (i.e., 190 lead SNPs with $P < 6.6 \times 10^{-9}$ and 96 SNPs with $6.6 \times 10^{-9} < P < 5 \times 10^{-8}$) by Warrington *et al*⁶, 273 SNPs were available in our dataset (**Table S2**).

Line 386-390: SNPs known to be associated with birthweight at $P < 5 \times 10^{-8}$ (i.e., 286 SNPs consisting 190 lead SNPs with $P < 6.6 \times 10^{-9}$ and 96 SNPs with $6.6 \times 10^{-9} < P < 5 \times 10^{-8}$) were extracted from a GWAS of birthweight by Warrington *et al*⁶. Out of the 286 SNPs, 273 (consisting 180 of the 190 lead SNPs and 93 of the 96 SNPs) were available in our dataset (**Table S2**).

Table S2. List of 273 SNPs included in the study.

The reviewer brings up an important point regarding fetal vs maternal effects, which is a topic of interest in perinatal genomics of complex traits. We studied SNPs irrespective of their cluster membership (SEM classifications: maternal/fetal/maternal-fetal same direction/maternal-fetal opposite direction/unclassified) after careful consideration of the challenge in delineating maternal from fetal effect. As emphasized by Warrington et al, those classifications, especially “maternal-effect” loci, are suggestive because of limitations of the study in statistical power and absence of paternal genotypes that lead to potential misclassification of parent-of-origin of alleles. Given this challenge in delineating maternal from fetal effects and the need for validation of the reported SEM classifications, we decided to include all loci because including only “fetal-related SNPs” may leave out several “unclassified SNPs” and some “maternal-only” SNPs that may have a small but undetected fetal effect operating via regulation of placental molecular traits. In a more recent study published during the review of our manuscript using a larger cohort and paternal genotypes that facilitated a more accurate phasing (Juliusdottir et al Nature Gen 2021), some SNPs in Warrington et al 2019’s suggested “maternal-only” category have also been found to have additional “fetal effect”, supporting their inclusion in our analysis. In fact, one of the advantages of a molecular regulatory study such as the one we performed is its contribution in advancing insights in distinguishing fetal effects operating through placental molecular traits from maternal effects. In this regard, our study found no QTL for loci that have shown “maternal-only effect” in both Warrington et al’s and the recent study. Overall, genetic regulation of fetal growth is extremely complex and unambiguous assignment of maternal vs fetal effect clusters is a work in progress. Hence, we have expanded the Discussion as follows:

Line 333-347: First, the maternal genome can indirectly influence placental molecular traits by altering the intrauterine environment. Some genetic loci with effects that vary by parent-of-origin have profound influence on placental development and function^{60,61} and birthweight⁵⁵. Understanding the genetic regulation of fetal growth is faced with the challenging task in delineating maternal genetic effects from fetal genetic effects, and in determining whether fetal effects depend on the variant’s parental origin. We did not consider suggested classifications of the GWAS SNPs via the fetal and maternal genome because cluster assignment remains “unclassified” for several birthweight GWAS loci, and unambiguous assignment of these classifications is a work in progress. For example, some SNPs previously classified to act only via the maternal genome have been found to have additional effect via the fetal genome in a recent larger study with fetal, maternal, and paternal genotypes that improved accuracy of genotype phasing⁵⁵. Although it is reassuring that the colocalized loci we identified did not overlap with variants that have previously been found to influence birthweight via the maternal, but not fetal, genome^{6,55}, our study cannot distinguish direct fetal genetic effects from indirect maternal genetic effects.

4. Birth weight is jointly determined by relative fetal growth and by gestational duration. Warrington et al 2019 et al adjusted analysis of birth weight for fetal sex and gestational duration. I searched the MS for the word “gestational” and could not find any reference to adjustment for gestational duration.

Could they please clarify if they simply analyzed crude birth weight? If so, this is a significant weakness. A 2500g birth weight would be normal at 37 weeks but small for gestational age at 41 weeks.

We fully agree at the importance of considering gestational age in studies of birthweight, and throughout our analyses, we used the covariate-adjusted summary statistics from Warrington et al's study. Gestational age was not mentioned in the previous version of our manuscript because our analysis was implemented on Warrington's GWAS summary statistics as input. We have now tested association between expression of the colocalized genes and birthweight (in response to reviewer 2's suggestion), and the analyses have been adjusted for fetal sex and gestational age (Line 227-243).

5. Lines 99-103. There are a series of proportions indicating overlap. But some overlap would be expected by the play of chance. It would be useful to know in each of the cases of overlap what the P value was for the observed versus the expected overlap based on the null hypothesis.

We have now added the overlap test p-values as follows:

Line 104: "... (vs. 0.04% expected genome-wide²⁵, $P=1.97 \times 10^{-54}$, hypergeometric test)"

Line 107: "... (vs. 0.2% expected genome-wide²⁵, $P < 10^{-300}$, hypergeometric test)"

Line 110: "... (14.3% vs. 3.1% expected genome-wide²⁵, $P=1.84 \times 10^{-11}$, hypergeometric test)"

Line 112: "... (vs. 18.3% genome-wide²⁵, $P=8.51 \times 10^{-19}$, hypergeometric test)"

6. Lines 140-142. Was the $P < 0.05$ corrected for a 5% FDR? Was the finding of 45 DNAm-eGene pairs greater than would have been predicted by the play of chance? How many pairs were $P < 0.05$ with a 5% FDR?

We have now added the overlap test p-values and FDR P-values as follows:

Line 153-156: "... ($P < 0.05$; FDR $P < 0.05$ for 13 DNAm-eGene pairs, < 0.1 for 36 DNAm-eGene pairs, < 0.25 for all 45 DNAm-eGene pairs) (Table S6). There was significant enrichment of DNAm-eGene pairs with causal directions (45/197 (22.8%) vs. 20.3% in a genome-wide dataset³⁰, $P=1.17 \times 10^{-2}$, hypergeometric test)."

7. Lines 151-154. Consistent predictions were observed using CIT and MR Steiger for three triplets. Is this more than would have been expected by the play of chance?

We have now added the overlap test p-values as follows:

Line 168: "...3/45 vs. 88/197, $P=2.96 \times 10^{-5}$, hypergeometric test.

8. Lines 194-202. External validation is crucial in demonstrating that the findings are correct and generalizable. Was the degree of overlap in the findings of the main study and the RICH study more than would be consistent with the play of chance? This should be spelled out in terms of the main findings, the observed overlap, the expected overlap based on the null hypothesis, and the P value for comparing observed versus expected based on the null hypothesis.

We have now added the overlap test p-values as follows:

Line 219: (vs. 0.2% genome-wide²⁵, $P=7.41 \times 10^{-169}$, hypergeometric test)

Line 223: (vs. 0.04% genome-wide²⁵, $P=5.40 \times 10^{-16}$, hypergeometric test)

Reviewer #2 (Remarks to the Author):

This manuscript performed a multi-omics “post-GWAS” approach to mining the intermediate markers of DNA methylation and gene expression to provide mechanistic insight into genetic SNPs associated with birthweight. Placenta tissue from a total of 71 births (+159 in replication cohort) for which imputed SNPs, Infinium EPIC array, and RNA-seq data (performed in previous studies) were all available. They started with 273 SNPs previously associated with birthweight and limited the analysis of mQTLs and eQTLs to 1 MB from each SNP, resulting in 197 mQTL-eQTL “triplets” mapping to 26 genes. They then performed 2 statistical tests of causality of directionality and an additional multi-trait colocalization test on the triplets. The following gene loci exhibited evidence for mQTLs with methylation causally influencing expression in placenta: WNT3A, CTDNEP1, and RANBP2. The following loci showed significant multi-trait colocalization: PLEKHA1, FES, CTDNEP1, and PRMT7. These results provide important functional insight into how birthweight GWAS SNPs may functionally interact with epigenetic mechanisms to regulate gene expression in placenta. Overall, the study appears to be well performed and described. However, there are a number of places where the approaches and populations in the study could be better described.

We thank the reviewer for their positive remark on our manuscript, and for suggesting areas for improvement. We have accommodated the requested changes in the revision (please see below).

1. *The authors admit that a limitation to their study was the lower number of placental samples for which RNA-seq was performed compared to EPIC. But the reason for this was not explained but important to include. For instance, did it have to do with RNA quality, budget, lack of overlap in existing studies, or some other type of selection? The concern would be to try to reduce the impacts of selection bias in the relatively small sample size of 71 participants for the discovery group.*

We have now clarified that resource (budget) was the reason RNAseq was performed in a subset of the samples. We have also added a new Table that shows a comparison of the characteristics of the subset of study participants for which placental eQTL analysis was performed using RNAseq with the larger set of participants for which placental mQTL analysis was performed using DNA methylation (Table S1). There was no significant difference in the characteristics of the study participants in the two sets, reassuring us that there is minimal impact of selection bias.

Line 85-92: A total of 291 placental samples with genotype and DNAm data were included in *cis*-mQTL analysis. Samples were obtained from pregnant women with self-identified Hispanic (n=97), White (n=74), Black (n=71), and Asian ethnicity (n=49); mean

age of 27.8 years; mean pre-pregnancy body mass index (BMI) of 25.1 kg/m²; and 46.5% female infants. A subset of 71 samples with genotype and RNAseq data (22 Hispanic, 21 White, 20 Black, 8 Asian) were included in *cis*-eQTL analysis. The characteristics of the study participants including maternal age, race/ethnicity, pre-pregnancy BMI, gestational duration, parity, and birthweight showed no significant difference between the participants with DNAm data and the subset with RNAseq data (**Table S1**).

Line 97-99: Further evaluation of the co-occurring eQTL and mQTL associations was performed in RICHs (n=159; 77.4% White; mean maternal age of 30.9 years; mean pre-pregnancy BMI of 26.6 kg/m²; 49.7% female infants (**Table S1**)).

Line 350-355: The size of our eQTL dataset is smaller than the mQTL dataset due to resource limitations, but is comparable to or larger than most tissue-specific datasets included in GTEx. Absence of significant difference in basic characteristics of the subset of study participants included in the eQTL analysis and those included in the mQTL analysis is reassuring that selection bias had minimal impact in our findings.

Table S1. Characteristics of the study participants included in the discovery and replication analyses.

2. While the study was based on multi-omics data that was individually published previously, the authors should provide information about study participants specifically used in this study, including variables such as age, gestational age, parity, maternal BMI, child sex, race/ethnicity, etc. This should be provided for both the discovery and replication cohorts.

Thank you for your suggestion. As mentioned in our response to comment #1, we have now included a Table and have described the study participants in the discovery and replication cohorts (Table S1).

3. The wording describing results was sometimes imprecise, particularly in the sections describing enrichments (lines 203-222). The terms “upregulated” and “downregulated” are meaningless without context. Using phrases such as “associated with birthweight” and other more precise language in this section would help improve clarity. Other places in the text could also use reminders to the reader that the associations were with birthweight.

We have expanded the description of the regulatory annotations to clarify the context within which these enrichments were derived from using phrases such as “in response to” and “relative to” as follows:

Line 254-262: Consistent with this finding, gene set enrichment analysis (GSEA) found significant enrichment of upregulated hallmark gene sets associated with low oxygen levels (oxidative stress), immune response, adipocyte differentiation (adipogenesis), development of skeletal muscle (myogenesis) and pancreatic beta cells (Table S16). Significant enrichment was also observed for differential gene expression in GTEx tissues (Bonferroni-adjusted $P < 0.05$), with the strongest enrichment for upregulation of the

gene set detected in subcutaneous adipose tissues relative to other GTEx tissues and the strongest enrichment for downregulation of the gene set detected in brain relative to other GTEx tissues (Fig. S2).

4. The “author contribution” section only describes the contributions of 3 out of 9 authors. While likely an inadvertent oversight, this is concerning if authors are included who did not contribute in a meaningful way.

We apologize for the oversight. We have updated the “author contribution section” as follows:

Line 525-530: Author contributions: FT-A conceived and designed this study and wrote the draft manuscript; XZ, SC, MO performed statistical analyses and visualizations; CL analyzed the replication dataset; MT reviewed functional annotations; TW contributed to data quality control; KH and JC contributed data; CJM and RW contributed data and samples. All authors contributed to interpretation of the results, reviewed the draft manuscript, provided critical intellectual content, and approved the final manuscript.

Reviewer #3 (Remarks to the Author):

The goal of this study is to provide functional mechanistic insights linking genetic variation to birthweight (BW). GWAS variants associated with BW were linked to methylation and gene expression data specific to the placenta. Findings suggest a causal role for a specific set of genes, previously linked to cardiometabolic, immune and hormonal regulation. Overall, the novelty of the approach, particularly the application to the placenta, makes this an interesting study that that will form the basis of several future investigations. The main problem with the data generated is the lack of any direct evidence that the identified associations are (i) specific in any way to the placenta and (ii) that they impact placental functioning in a manner that might explain variation in BW.

Thank you for the thoughtful feedback and positive remark about the novelty of our work and potential future impact. We have addressed the specific comments provided, and included a response below. To further evaluate whether the evidence of colocalization for the four loci identified is specific to placenta, we assessed multi-trait colocalization (birthweight, gene expression, DNA methylation) in blood from two sources, and found no significant evidence of colocalization for any of the four loci. We have also performed two-trait colocalization between birthweight and gene expression available for GTEx tissues (despite limitation that multi-trait colocalization cannot be done due to lack of DNA methylation data). Overall, the results suggest that PLEKHA1 and CTDNEP1 are likely to be placenta-specific. We added the results and discussion as shown below:

Line 210-214: In blood “...no significant ($PPA \geq 0.8$) multi-trait colocalization was found. There was suggestive evidence of colocalization between *PLEKHA1* gene expression in blood and birthweight ($PPA = 0.76$) (**Table S8**). To expand our assessment to non-

placenta tissues (with the caveat that methylation data are not available), we tested colocalization for gene expression and the four birthweight loci in 48 GTEx tissues using coloc²³⁵. Whereas no significant colocalization was found for *PLEKHA1* and *CTDNEP1*, evidence of colocalization was found for *PRMT7* and *FES* in some tissues (**Table S9**)."

Line 325-332: The colocalized genes we identified are broadly expressed in several tissues, but our evaluation in blood-derived eQTL from GTEx and mQTL from ARIES found no statistically significant evidence of multi-trait colocalization. Moreover, we did not observe eQTL-birthweight colocalization for *PLEKHA1* and *CTDNEP1* in any GTEx tissue. It is possible that the variants influence birthweight via the effector genes acting primarily in placenta, whereas the same variants may influence cardiometabolic traits in later life via other mechanisms. Multiple tissues with mQTL and eQTL data derived from identical samples will be critical to confirm this.

To assess potential direct impact of the associations on placental function, we have now tested associations of colocalized genes' placental expression with placental weight in the NICHD Fetal Growth Studies (placental weight is unavailable in RICHHS cohort, so was not tested). As suggested by reviewer #1, we have also tested associations of colocalized genes' placental expression with birth weight and the results are presented in Table S11 and results section (see text below). We have also expanded the discussion section on function of the genes in placental development and function, strengthening the evidence on potential placental mechanisms impacting fetal growth.

Line 302-308: We observed that genetic variants associated with higher placental *HTRA1* expression have birthweight-lowering effect. Similarly, placental expression of *htra1* is significantly increased in preeclampsia, a pregnancy complication associated with low birthweight⁵¹. It has been suggested that *htra1* over-expression in placenta may disrupt microtubule organization, attenuated cell motility, and endothelial dysfunction^{52,53}, placental trophoblast invasion-related features critical in fetal growth.

Line 227-243: Association between colocalized gene expression and birthweight. We assessed whether placental expression of the colocalized genes was associated with birthweight using linear regression adjusted for fetal sex and gestational age. *PLEKHA1* and *HTRA1* expression levels were significantly inversely associated with birthweight in the RICHHS dataset and the association strengthened in meta-analyses of the results from RICHHS and NICHD Fetal Growth Studies. *HTRA1* expression showed suggestive inverse association with placental weight in the NICHD Fetal Growth Studies ($P=0.06$). Moreover, *PRMT7* and *CTDNEP1* expression levels showed directionally consistent inverse associations with birthweight in both datasets, although not reaching statistical significance (**Table S11**). The inverse direction of association between these four genes and birthweight concurred with the eQTL findings such that the alleles with birthweight-

lowering effect also had gene expression-increasing effect in placenta. It is worth noting that gene expression levels are attributed to both genetic and environmental factors, and traits such as birthweight can in turn drive gene expression differences³⁶. As a result, the relationship between *cis*-regulated gene expression and a trait may not be accurately mirrored by the relationship between measured gene expression and a trait. More accurate inference of the portion of gene expression, partitioned into genetic and non-genetic components, causally associated with birthweight warrants further investigation in larger well-characterized cohorts.

Strengths

- 1) *Very interesting study design and an important hypothesis that has largely been ignored to date. The three-step approach to build evidence linking specific BW associated SNPs to placental expression and methylation is really nice.*
- 2) *Reasonably large sample size of 291 samples (with ethnic diversity) with both SNP and methylation data, some with gene expression data. Overall, the sample is adequate to reveal some novel findings, but is far from conclusive*
- 3) *Use of independent dataset provides further support for the robust nature of the findings is good*
- 4) *Several of the study limitations are highlighted*

We appreciate the reviewer's positive remarks on the value of our hypothesis, approach, and findings.

Limitations

- 1) *Recent studies suggest a contribution of both indirect maternal and direct fetal genetic effects on BW. It is now clear that performing GWAS solely on fetal genetics precludes full resolution of the origin of the identified genetic effects. In other words, maternal genetic factors are also likely very important in regulation of placental structure and function, and therefore, BW*

We agree that distinguishing fetal vs maternal effects is a topic of growing interest in perinatal genomics of complex traits. We have expanded the discussion remarking the value of the maternal genome in placental function and fetal growth. Please also see the revisions we made in response to comment #3 below.

Line 333-347: First, the maternal genome can indirectly influence placental molecular traits by altering the intrauterine environment. Some genetic loci with effects that vary by parent-of-origin have profound influence on placental development and function^{60,61} and birthweight⁵⁵. Understanding the genetic regulation of fetal growth is faced with the challenging task in delineating maternal genetic effects from fetal genetic effects, and in determining whether fetal effects depend on the variant's parental origin. We did not consider suggested classifications of the GWAS SNPs via the fetal and maternal genome because cluster assignment remains "unclassified" for several birthweight GWAS loci, and unambiguous assignment of these classifications is a work in progress. For example, some SNPs previously classified to act only via the maternal genome have been found to have additional effect via the fetal genome in a recent larger study with

fetal, maternal, and paternal genotypes that improved accuracy of genotype phasing⁵⁵. Although it is reassuring that the colocalized loci we identified did not overlap with variants that have previously been found to influence birthweight via the maternal, but not fetal, genome^{6,55}, our study cannot distinguish direct fetal genetic effects from indirect maternal genetic effects.

2) *The most recent examination of the genetic influences on BW identified a range of biological pathways that contribute to this phenotype including variants influencing blood pressure, glycemic traits and height (amongst others). Few of these would have any link to placental function but should not be disregarded as important determinants of BW*

We have now highlighted the influence of cardiometabolic trait-related genetic variants on birthweight in the discussion, as follows:

Line 317-325: Accumulating evidence shows considerable shared genetic overlap between fetal growth and cardiometabolic traits in later life^{6,55-57}. Polygenic variants in the fetal genome associated with higher glucose and blood pressure have a birthweight-lowering effect. In contrast, polygenic variants in the maternal genome associated with higher glucose have a birthweight-raising effect possibly due to increased cross-placental transfer of glucose that induces insulin secretion and promotes fetal growth^{55,58,59}. The birthweight-associated SNPs with colocalized effect on placental gene expression and methylation in our study have previously been associated with adult cardiometabolic phenotypes including blood pressure, type 2 diabetes, coronary artery disease, and height^{6,55}.

3) *More information is needed on the starting 286 BW associated SNPs. Warrington et al, identified 190 independent SNP associations derived from 146 offspring own BW and 72 maternal effect SNPs. How does this impact the interpretation of the current study where only offspring genotype was considered?*

We have now clarified in the Results and Methods sections that the 286 SNPs consist of all SNPs in Warrington et al 2019 study passing the GWAS significance threshold of $P < 5 \times 10^{-8}$ (i.e., 190 lead SNPs (from a total of 209 SNPs) with $P < 6.6 \times 10^{-9}$ and 96 SNPs with $6.6 \times 10^{-9} < P < 5 \times 10^{-8}$). [Note that in a look-up of the 96 loci in a larger study published while our manuscript was under review (*Nat Genet* 53, 1135-1142 (2021)), we observed that more than 1/3rd were found to be genome-wide significant at class-based significance thresholds, strengthening our decision to include all loci passing the GWAS threshold]. We have now added the list of analyzed SNPs in a Table S2 and clarified the Results and Methods sections as follows:

Line 91-93: Out of 286 single nucleotide polymorphisms (SNPs) found to be associated with birthweight at $P < 5 \times 10^{-8}$ (i.e., 190 lead SNPs with $P < 6.6 \times 10^{-9}$ and 96 SNPs with $6.6 \times 10^{-9} < P < 5 \times 10^{-8}$) by Warrington et al⁶, 273 SNPs were available in our dataset (**Table S2**).

Line 386-390: SNPs known to be associated with birthweight at $P < 5 \times 10^{-8}$ (i.e., 286 SNPs consisting 190 lead SNPs with $P < 6.6 \times 10^{-9}$ and 96 SNPs with $6.6 \times 10^{-9} < P < 5 \times 10^{-8}$) were extracted from a GWAS of birthweight by Warrington *et al*⁶. Out of the 286 SNPs, 273 (consisting 180 of the 190 lead SNPs and 93 of the 96 SNPs) were available in our dataset (**Table S2**).

We agree with the reviewer that fetal and maternal genetic effect is a topic of growing interest. We studied SNPs irrespective of their cluster membership (SEM classifications: maternal/fetal/maternal-fetal same direction/maternal-fetal opposite direction/unclassified) after careful consideration of the challenge in delineating maternal from fetal effect. As emphasized by Warrington *et al*, those classifications, especially “maternal-effect” loci, are suggestive because of limitations of the study in statistical power and absence of paternal genotypes that lead to potential misclassification of parent-of-origin of alleles. Given this challenge in delineating maternal from fetal effects and the need for validation of the reported SEM classifications, we decided to include all loci because including only “fetal-related SNPs” may leave out several “unclassified SNPs” and some “maternal-only” SNPs that may have a small but undetected fetal effect operating via regulation of placental molecular traits. In a more recent study published during the review of our manuscript using a larger cohort and paternal genotypes that facilitated a more accurate phasing (Juliusdottir *et al* *Nature Gen* 2021), some SNPs in Warrington *et al* 2019’s suggested “maternal-only” category have also been found to have additional “fetal effect”, supporting their inclusion in our analysis. In fact, one of the advantages of a molecular regulatory study such as the one we performed is its contribution in advancing insights in distinguishing fetal effects operating through placental molecular traits from maternal effects. In this regard, our study found no QTL for loci that have shown “maternal-only effect” in both Warrington *et al*’s and the recent study. Overall, genetic regulation of fetal growth is extremely complex and unambiguous assignment of maternal vs fetal effect clusters is a work in progress. We have expanded the Discussion as follows:

Line 324-338: First, the maternal genome can indirectly influence placental molecular traits by altering the intrauterine environment. Some genetic loci with effects that vary by parent-of-origin have profound influence on placental development and function^{60,61} and birthweight⁵⁵. Understanding the genetic regulation of fetal growth is faced with the challenging task in delineating maternal genetic effects from fetal genetic effects, and in determining whether fetal effects depend on the variant’s parental origin. We did not consider suggested classifications of the GWAS SNPs via the fetal and maternal genome because cluster assignment remains “unclassified” for several birthweight GWAS loci, and unambiguous assignment of these classifications is a work in progress. For example, some SNPs previously classified to act only via the maternal genome have been found to have additional effect via the fetal genome in a recent larger study with fetal, maternal, and paternal genotypes that improved accuracy of genotype phasing⁵⁵. Although it is reassuring that the colocalized loci we identified did not overlap with variants that have previously been found to influence birthweight via the maternal, but

not fetal, genome^{6,55}, our study cannot distinguish direct fetal genetic effects from indirect maternal genetic effects.

4) 273 SNPs were available in the placental dataset but there is no information given on how many of the variable genotypes were actually present and at what frequency. This should be included as a table

We have added Table S2 in the Supplement with a list of the 273 SNPs and their genotype allele frequency (Table S2).

Table S2. List of 273 SNPs included in the study.

5) If not already included, it is important that the specific SNP, CpG and genes of interest be included in the current study, particularly the distance from sites of interest to the genes linked to them.

All co-occurring eQTL and mQTL SNPs, genes and CpGs have been included in Tables S3 and S4. We have also updated Table 1 by adding base-pair distance between SNP, CpG, and gene for the colocalized loci.

6) GTEx data from non-placental tissues indicates a likely large component of these tissues to the observed effects on BW. This needs to be discussed more

We have expanded the discussion on this.

Line 317-332: Accumulating evidence shows considerable shared genetic overlap between fetal growth and cardiometabolic traits in later life^{6,59-61}. Polygenic variants in the fetal genome associated with higher glucose and blood pressure have a birthweight-lowering effect. In contrast, polygenic variants in the maternal genome associated with higher glucose have a birthweight-raising effect possibly due to increased cross-placental transfer of glucose that induces insulin secretion and promotes fetal growth^{59,62,63}. The birthweight-associated SNPs with colocalized effect on placental gene expression and methylation in our study have previously been associated with adult cardiometabolic phenotypes including blood pressure, type 2 diabetes, coronary artery disease, and height^{6,59}. The colocalized genes we identified are broadly expressed in several tissues, but our evaluation in blood-derived eQTL from GTEx and mQTL from ARIES found no statistically significant evidence of multi-trait colocalization. We also did not find significant eQTL-birthweight colocalization for *PLEKHA1* and *CTDNEP1* in any GTEx tissue. It is possible that the variants influence birthweight via the effector genes acting primarily in placenta, whereas the same variants may influence cardiometabolic traits in later life via other mechanisms. Multiple tissues with mQTL and eQTL data derived from identical samples will be critical to confirm this.

7) Little data on the localisation of expression of genes of interest within the placenta is shown.

We have added a Supplementary Table (Table S7) with summary description of gene expression, tissue specificity, predicted location of protein expression, and molecular and protein function; and a Supplementary Figure (Fig. S1) on number of DNA methylation CpG sites per regulatory annotation. We have added the following text:

Line 199-203: The colocalized genes are broadly expressed in several tissues, with location of protein expression predicted as intracellular, and molecular functions including lipid-binding (*PLEKHA1*), growth factor binding (*HTRA1*), chromatin regulator, kinase, and transferase (*PRMT7*, *FES*) (**Table S7**). The colocalized DNAm sites are predominantly annotated to genomic regions with intron, promoter, and CpG shore features (**Fig. S1**).

8) Very little information is given on the relative localisation of specific CpGs, SNPs or gene elements (promoters, exons etc), particularly those listed in Table 1.

We have updated Table 1, adding the position of the SNPs and CpG sites, and relation of the CpGs with the genes (body, exon, 5'UTR, etc), and annotated CpGs with promoter-associated regulatory features. Please also see our response to comment #7.

9) Minor points: Although true a decade ago, it is disputable whether the placenta remains understudied in terms of genomics. Despite this, its exclusion from important international initiative such as GTEx has been unfortunate.

Thank you, we have updated the sentence in the abstract as follows:

Line 26-27, Abstract: “Although the placenta is critical to fetal development and later life health, it has not been integrated into largescale functional genomics initiatives,…”

Reviewers' Comments:

Reviewer #1:

Remarks to the Author:

I have carefully reviewed the authors' response to the reviewers' comments and I have reviewed the revised MS. I think the authors have addressed the reviews well and I have no further comments.

Reviewer #2:

Remarks to the Author:

The authors have effectively responded to prior concerns and the manuscript is significantly improved.